

# Enhanced chlorophyll-*a* concentration in the wake of Sable Island, eastern Canada, revealed by two decades of satellite observations: a response to grey seal population dynamics?

Emmanuel Devred[1], Andrea Hilborn[1], and Cornelia den Heyer[1]

[1]Bedford Institute of Oceanography, 1 Challenger Drive, Dartmouth, NS, B2Y 4A2, Canada

**Correspondence:** Emmanuel Devred (Emmanuel.Devred@dfo-mpo.gc.ca)

**Abstract.** Elevated surface chlorophyll-*a* concentration, an index of phytoplankton biomass, has been previously observed and documented by remote sensing in the waters to the southwest of Sable Island (SI) on the Scotian Shelf in eastern Canada. Here, we present a detailed analysis of this phenomenon using a 20-year time series of satellite-derived chlorophyll-*a* concentration (chl-*a*), paired with information on the particle backscattering coefficient at 443 nm ($b_{bp}(443)$) and the detritus/gelbstoff

absorption coefficient at 443 nm ($a_{dg}(443)$) in an attempt to explain the possible mechanisms that lead to the increase in surface biomass in the surroundings of SI. We compared the seasonal cycle, climatology and trends of surface waters near SI to two control regions located both upstream and downstream of the island, away from terrigenous inputs. Application of the self-organizing maps approach (SOMs) to the time series of satellite-derived chl-*a* over the Scotian Shelf revealed the annual spatio-temporal patterns around SI and, in particular, persistently high phytoplankton biomass during winter and spring in the

leeward side of SI, a phenomenon that is not observed in the control boxes. Time series analysis of the satellite archive evidenced a long-term increase in chl-*a* and $a_{dg}(443)$, and a long-term decrease in $b_{bp}(443)$ in all regions. In the close vicinity of SI, the increase of chl-*a* and $a_{dg}(443)$ during the winter months occurred at a rate twice that of the ones observed in the control boxes. In addition to the increase of the chl-*a* and $a_{dg}(443)$ within the plume southward of SI, the surface area of the plume itself has also expanded by a factor of five over the last 20 years. While the island mass effect (IME) is certainly contributing

to the enhanced biomass around SI, we hypothesize that the large increase in chl-*a* over the last 20 years is due to an injection of nutrients by the island's grey seal colony, which has increased by ∼300% over the last twenty years. The contribution of nutrients from seals may sustain high phytoplankton biomass at a time of year when it is usually low. A conceptual model was developed to describe the annual variation of seal abundance on SI and estimate the standing stock of chl-*a* concentration that can be sustained by the release of nitrogen. Comparison between satellite observations and model simulations showed a very

good agreement between the seal population increase on SI during the breeding season and the phytoplankton biomass increase during the winter. In addition, the 20-year satellite-derived trend in chlorophyll-*a* concentration showed a good agreement with the increasing trend in seal population on SI during the same time period. The satellite data analysis supports the concept of top-down control of marine mammals over lower trophic levels through a fertilisation mechanism, although these results could not be confirmed without *in situ* measurements for ground truthing. Our findings challenge the idea that the IME is restricted

to islands with strong bathymetric slope located in oligotrophic waters of mid-latitudes and tropics, and demonstrate that en-





hanced marine production can occur in other oceanic regions, with potentially substantial implications for conservation and fisheries.

## 1 Introduction

Increased phytoplankton production around islands, due to the island mass effect (IME), has been documented in many studies since the development of the concept (Doty and Oguri, 1956). This phenomenon, often occurring leeward of islands, results from hydrodynamic forcing (e.g., flow disturbance) that induces mixing of the water column, and provision of nutrients originating from benthic processes (Signorini et al., 1999), or land drainage (Dandonneau and Charpy, 1985) to the well-lit upper layer of the ocean (Heywood et al., 1990) in turn sustaining enhanced phytoplankton production (e.g., Gilmartin and Revelante,

1974; Gove et al., 2016; Messié et al., 2020; James et al., 2020). Evidence of the IME has been documented in the proximity of islands located in oligotrophic oceanic basins (Dandonneau and Charpy, 1985; Gove et al., 2016; Messié et al., 2020) and in the sub-Antarctic environment (Boden, 1988). Ocean colour remote sensing (OCRS) has proven to be a useful tool to observe increased primary production around islands and study its spatio-temporal variation given the fine spatial and temporal scales achievable in comparison to *in situ* sampling (e.g., Signorini et al., 1999; Gove et al., 2016; Martinez et al., 2018; Martinez

and Maamaatuaiahutapu, 2004; James et al., 2020). For the first time, to our knowledge, an increase in production due to the IME around an island located in the mesotrophic waters of the eastern Canadian continental shelf, Sable Island (SI), is investigated using OCRS. We further discuss possible mechanisms that explain the high phytoplankton biomass and in particular, we consider a previously neglected source of nutrient entrainment into the local ocean; the presence of the world's largest colony of grey seals (*Halichoerus grypus*). SI supports the world's largest breeding population of grey seals. The breeding colony on

Sable Island has grown rapidly since the 1960s when just a few thousand pups were born on the island (Bowen et al., 2007; den Heyer et al., 2020). The population associated with the SI breeding colony is now over 300,000 individuals (Hammill et al., 2017), and the impact of this population on prey species has been focus of much research (e.g. Trzcinski et al., 2006; O'Boyle and Sinclair, 2012; Hammill et al., 2014; Neuenhoff et al., 2019). Recently the grey seal breeding colony has also been shown to contribute to the ecology of the island by fertilizing vegetation that supports a population of feral horses (McLoughlin et al.,

2016). Previous studies have demonstrated the impact of marine mammals on the supply of nutrients through turbulent mixing (Kanwisher and Ridgway, 1983), by direct release of nitrogen in the marine environment (Roman and McCarthy, 2010; Laver et al., 2012; Mccauley et al., 2012; Wing et al., 2014) and atmospheric deposition (Theobald et al., 2006).

The Ocean Colour Climate Change Initiative (OC-CCI) satellite time series over the period 1998-2018 was used to investigate the dynamics of chlorophyll-*a* concentration (chl-*a*), absorption by dissolved organic matter, and the backscattering

coefficient (an index of biogenic and mineral particles, Boss et al., 2004; Slade and Boss, 2015) around SI and over a portion of the Scotian Shelf. The first objective of the study was to demonstrate and quantify the higher phytoplankton biomass





around SI and evidence the IME. Given that the time series of satellite observations span over two decades, we analysed the climatologies and seasonal cycles of several sub-regions of the Scotian Shelf, including SI, using the artificial neural network technique of self-organizing maps (SOMs) to isolate recurring patterns of chl-*a* and we investigated decadal trends of the three

remote sensing parameters. While the information provided by satellites remains limited to marine components that are optically active, the second objective of the study was to examine the possible mechanisms causing such an increase of biomass in space and time around the island. To that point, while most satellite-based studies of the relationship of IME to production have been limited to chlorophyll-*a* concentration, we also considered variations in absorption by yellow substances and detritus to help track possible runoff from the island and significant sources of DOM, and variation in the backscattering coefficient,

which indicates the presence of mineral particles in the water column near the island due to re-suspension. The first part of the manuscript presents the area of interest, data and methodology, with the results and discussion section organised to answer the following questions:

  – Is there an island mass effect, and if so, what is its spatio-temporal distribution?

  – Are there decadal trends in the satellite-derived properties around SI?

– What is(are) the possible mechanism(s) that can explain the trends?

## 2 Data and Methods

### 2.1 Study Area

The Scotian Shelf bioregion is a significant marine shelf region due to its biological richness and diversity (Ward-Paige and Bundy, 2016). The influence of the warm gulf stream makes the area home to marine species normally found further south,

and it is used as spawning and nursery grounds by many fish species. The mean surface circulation consists of a persistent current flowing to the south-west, trapped by the coast (the Nova Scotia Current, NSC), and a parallel cold current flowing to the south-west (Labrador Current, LC) along the shelf break (Smith and Schwing, 1991; Loder et al., 1997) (Figure 1). The Scotian Shelf is bounded to the east by the Laurentian Channel, to the west by the Northeast Channel, and to the south by the continental shelf break oriented parallel to coastal Nova Scotia. SI is the only island on the shelf, emerging centrally adjacent

to the shelf break from Sable Island Bank (see Figure 1); most of the bank is an Ecologically and Biologically Significant Area (EBSA) due to its high abundance of groundfish and persistently high chlorophyll-*a* concentration (King et al., 2016). While the average depth of the Scotian Shelf is 90 m, there is complex circulation over banks and basins as deep as 300 m, and Sable Island Bank lies at a depth of approximately 60 m. SI itself has a surface area of only 34 km² (about 49 km in length with large interannual variation, and 1.25 km at its widest point) and is composed of sand and partially vegetated by Marram

grass (*Ammophilia breviligulata*), other grasses, forbs and shrubs, and as a result, undergoes little surface runoff. Precipitation directly recharges a freshwater aquifer that runs the length of the island maintained by hydrostatic pressure (Kennedy et al., 2014). SI is home to a population of feral horses in addition to hosting the largest breeding colony of grey seals (*Halichoerus grypus*) in the world notably in winter, when seals gather during the breeding season that spans from December to February.





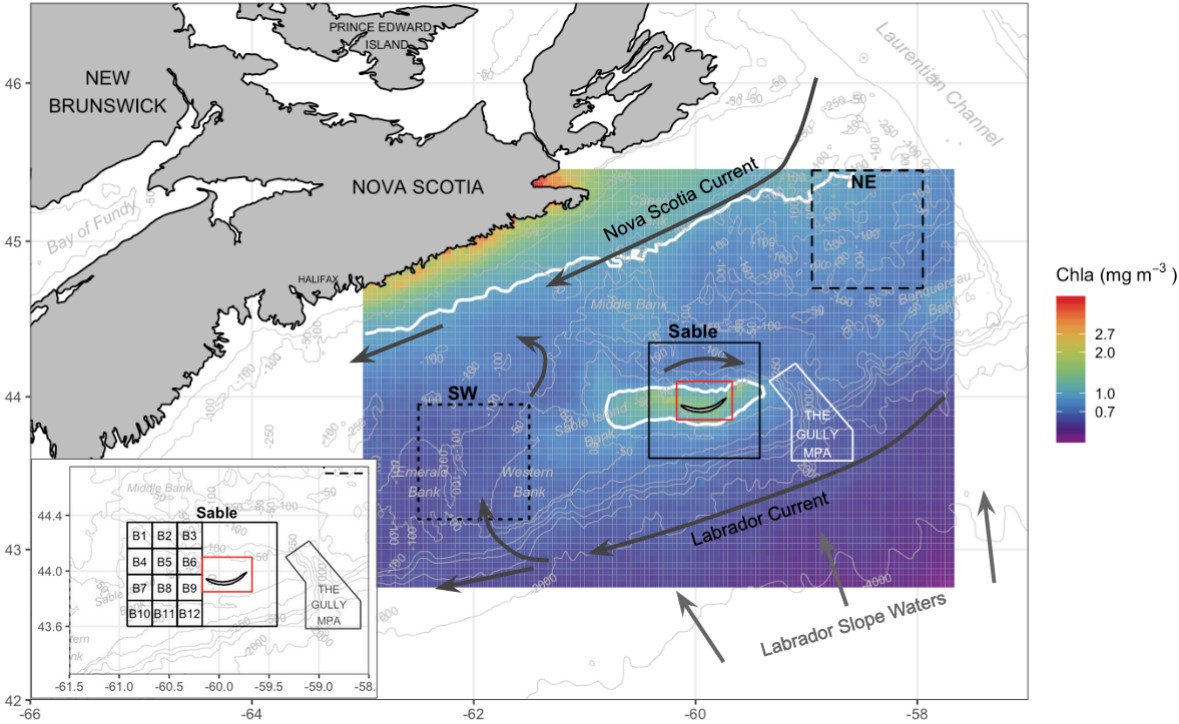

**Figure 1.** Map of the Scotian Shelf showing bathymetric features (grey lines), main hydrodynamic circulation (dark arrows) and the three regions used for analysis (solid black line is SI; short-dashed SW; long-dashed NE). Pixels in the red box were removed from analysis. The inset shows 12 small boxes adjacent to Sable Island (see Sect. 2.4). The mean chl-*a* over the period 1998-2018 is shown in the background, with white contours delineating 1.0 mg m$^{-3}$.

While the focus of this study is the marine region adjacent to SI, data were also analysed for two control boxes, which were

arbitrary selected near the island to evidence the unique increase in biomass around SI (Figure 1) and to ensure that the findings around SI were the results of local processes and not the reflection of a general pattern occurring over the entire Scotian Shelf. One box was located to the Northeast (i.e., upstream and referred to as "NE" with a mean depth of 111 m) and the second one was located to the Southwest (downstream and referred to as "SW" with a mean depth of 83 m) of SI. These boxes were located far enough from SI and the continent to avoid influences of terrigenous runoff. A large box centered on Sable Island

(referred to as "SI") of the same geographic size to NE and SW was defined to investigate the possible IME, and a grid made up of 12 small boxes to the west of the island was defined to infer the spatial extension of the SI plume (Figure 1, inset) without subjective delineation.



## 2.2 Satellite and environmental datasets

Satellite data were downloaded from the Ocean Colour Climate Change Initiative (OC-CCI, https://rsg.pml.ac.uk/thredds/catalog-
cci.html), a project providing validated, inter-sensor calibrated and error-characterized Essential Climate Variables (ECVs)
from several satellite sensors (OC-CCI dataset v4.0; Sathyendranath et al. (2019, 2020)). The dataset consists of merged prod-
ucts from the Sea-viewing Wide Field-of-View Sensor (SeaWiFS, 1997 - 2010), MEdium-Resolution Imaging Spectrometer
(MERIS, 2002 - 2012), MODerate resolution Imaging Spectroradiometer on Aqua satellite (MODISA, 2002 - present) and
Visible Infrared Imaging Radiometer Suite sensors (VIIRS, 2012 - present). For this study, 8-day 4km-resolution composites
of chlorophyll-$a$ (mg m$^{-3}$)), absorption coefficient of detrital and dissolved organic matter (DOM) at 443 nm ($a_{dg}(443)$, m$^{-1}$)
and total backscattering from particulate matter at 443 nm ($b_{bp}(443)$, m$^{-1}$) were extracted from the global dataset for a region
bounded by 42.7 - 45.45° N, and 57.6 - 63.0° W. The chl-$a$ was produced by the OC-CCI from remote-sensing reflectance
(Rrs) using the OC3 algorithm (see Jackson et al., 2020, for details), dependent on water class membership, where Rrs are
atmospherically corrected using the SeaWiFS Data Analysis System (SeaDAS 7.3) and POLYMER software (Steinmetz et al.,
2011). Inherent optical properties (IOPs) such as $a_{dg}(443)$ and $b_{bp}(443)$ are calculated at SeaWiFS wavelengths from Rrs using
the Quasi-Analytical Algorithm (Lee et al., 2005). For the SI box, a small rectangle containing the island and submerged shal-
lows was masked from all calculations (shown as the red box in Figure 1) to reduce the impact of pixels potentially influenced
by resuspended sediment or bottom reflectance. Further, large coccolithophore blooms occasionally occur in the surrounding
waters of SI, which impact the magnitude of $b_{bp}(443)$. To remove this effect, 8-day composite images with $b_{bp}(443)$ several
orders of magnitude greater than the annual mean were discarded for the period of June 25 through August 11, 2003, and June
25 through August 3, 2010. Lastly, 24 images with erroneous data were identified and removed during the SOM processing
(see section 2.4). The resulting remote sensing dataset used in this study consisted of 958 chl-$a$ and $a_{dg}(443)$, and 946 $b_{bp}(443)$
Level 3 8-day composites spanning years 1998-2018.

The 5-day composite NASA OSCAR sea-surface velocity dataset (http://www.oscar.noaa.gov/) at $1/3°$ resolution were
downloaded from the NOAA ERDDAP server (https://coastwatch.pfeg.noaa.gov/erddap, dataset ID: jplOscar_LonPM180)
for the corresponding time period to the satellite observations. The zonal (u in m s$^{-1}$) and meridional (v in m s$^{-1}$) compo-
nents of the sea-surface velocity field were further averaged into monthly and annual fields to examine prevailing current
speed and direction. We also examined the meteorological data over the period 1998-2018, including daily wind and pre-
cipitation, from an Environment and Climate Change Canada station located on Sable Island (ID 8204700, https://climate-
change.canada.ca/climate-data/#/daily-climate-data ). Bathymetry was acquired from NOAA ETOPO1 and used to calculate
chl-$a$ standing stocks (Section 2.5.2).

All analysis were performed using R statistical software (R Core Team, 2020).

## 2.3 Climatology and trends of satellite chl-$a$, $a_{dg}(443)$ and $b_{bp}(443)$

Eight-day climatologies of chl-$a$, $a_{dg}(443)$ and $b_{bp}(443)$ for Si and the two control boxes (i.e., SW and NE) were calculated
as the average of all pixels of all years within a given 8-day period and box (Figure 2). Pixels with values outside of the 1.5





interquartile range (IQR) were removed and, following that step, boxes with less than 30% of spatial coverage for a given 8-day period were discarded to ensure the highest quality of the data and prevent skewing of results by outliers. In addition, 8-day climatolgies were computed for a total of 30 small boxes ($0.2° \times 0.2°$) organised around SI to identify in an objective manner the region of influence of the IME. A total of 18 small boxes were not used for further analysis because they showed

a similar chl-*a* climatology than the control boxes, meaning that they were not impacted by the IME, however, these boxes helped delineated the region impacted by the IME. The analysis focused on the 12 remaining boxes to help understand and quantify the variability in chl-*a*, $a_{dg}(443)$ and $b_{bp}(443)$ southwestward of SI (Figure 1, inset). In addition to the eight-day time binning, seasonal composites of chl-*a*, $a_{dg}(443)$ and $b_{bp}(443)$ were computed as the arithmetic mean of images during winter, spring, summer and fall for the SW and NE boxes, and a plume located southwest of Sable Island (i.e., SOM5, see section

2.4 for details). Winter, spring, summer and fall were defined as the months of December to February, March to May, June to August, and September to November, respectively. The same quality control used for the 8-day climatology calculations were also implemented here. The seasonal binning was favoured against the 8-day composite for the trend analysis as it removes missing data in the time series. The annual and seasonal long-term trends were computed as the slope of the linear least-squares regression for chl-*a*, $a_{dg}(443)$ and $b_{bp}(443)$.

While the above time series and trend analysis are useful to quantify changes in the bio-optical properties around SI, we selected a different approach to bin the satellite data for comparison with model simulations (see section 2.5.2). The seasonal mean satellite data were averaged into 5-year time periods, namely, 1999-2003 (P1), 2004-2008 (P2), 2009-2013 (P3) and 2014-2018 (P4), to obtain detailed maps of the chl-*a* in the SOM5 without gaps (i.e., missing pixels). Note that the year 1998 was discarded from this analysis to have all periods equal to 5 years exactly. These maps were used to calculate chl-*a* standing

stocks from ocean colour satellite for comparison with chl-*a* standing stocks derived from seal nitrogen excretion (see section 2.5.2). The chl-*a* standing stocks independent of possible seal influence during P1 were computed by subtracting the chl-*a* standing stocks due to seal N excretion from the satellite observations and referred to as P1N. This initial values was linearly extrapolated to the three other periods (i.e., P2, P3 and P4) by applying the slope of the chl-*a* linear trends derived in the SW box, which is assumed to be too far from the island to be influenced by seal fertilisation. The P2N, P3N and P4N chl-*a* standing

stocks obtained using this method provided the increase in chl-*a* standing stocks due to climate forcing without any influence from the presence of seals on SI. Note that in November of 2011, an unusual and intense phytoplankton bloom occurred southward of SI, partially covering the SOM5 region but not connected to SI; this bloom that lasted almost the entire month was removed from the analysis as it contaminated the background chl-*a* and was not a result of the IME.

## 2.4 Chlorophyll-*a* concentration spatio-temporal variation using self-organising maps (SOMs)

Self-organizing map methods were applied to the satellite-derived chl-*a* dataset over the entire Scotian Shelf in order to quantify recurring spatial patterns in an unsupervised manner. The basic principle of a SOM is to map high-dimension data, in our case a three-dimensional dataset (i.e. two dimensions in space and one in time), to a 2-dimensional array of nodes.The *kohonen* R package (Wehrens and Buydens, 2007; Wehrens and Kruisselbrink, 2018) was applied as an unsupervised SOM method to the 20-year dataset of OC-CCI chl-*a* for a region bounded by -63 to -57.5°E and 42.8 to 45.5°N (Figure 3). A 3x3 hexagonal

**Figure 2.** Mean value per week of chl-*a* (a), $a_{dg}(443)$ (b) and $b_{bp}(443)$ (c) for the regions on the Scotian Shelf (see Figure 1). The three large regions are noted by black lines (SW is dotted, SB is solid, and NE is dashed) and the 12 small boxes are coloured.

layout SOM, i.e., nine nodes, was selected after trials of multiple layout configurations, as it provided low mean node distance, enough nodes to represent distinct spatial patterns of chl-*a* and avoid redundancy, each node accounting for a distinct general pattern across the time series (Figure 3a). Each image retains a measure of fit (i.e. resemblance) to the centroid of the node it



is assigned to (referred to as "node distance"), with a larger node distance indicating poorer membership. The final result has each node (i.e., chl-*a* map) representing a particular spatial pattern consisting of the distance-weighted average of its assigned

images, with similar patterns located nearer each other in the final SOM layout. This is an iterative procedure that should run until the overall distance of the entire system is minimum; in this case, a total of 500 iterations for each SOM trial was sufficient for the mean node distance to converge and for all individual node distances to stabilize to a similar value (0.0042, arbitrary units). Prior to processing, the daily chl-*a* images were log-transformed and arranged as a spatio-temporal matrix consisting of one image per row. Images with less than 10% spatial coverage were excluded from the analysis. Temporal gaps

were not accounted for, as images are assigned to the node array individually with no dependence on position in the time series. Further, 24 chl-*a* images were identified as having persistently high node distance through all trials (e.g. did not fit well with any node, regardless of array size). When visually inspected, these images displayed heterogeneous and often unrealistic chl-*a* values (e.g., very high); these images and corresponding $a_{dg}(443)$ and $b_{bp}(443)$ were removed from the dataset for all analysis including the SOM, climatology computation and time series analysis. The temporal occurrence of each node was examined,

and the progression of spatial patterns in the SOM space was calculated using the mean position (centroid) of the nodes in a given month (Figure 3b) as in (Richardson et al., 2003). Additionally, given that we are interested in the elevated chl-*a* near Sable Island in Winter, the SOM results were used to delineate a region with chl-*a* exceeding $1\,\mathrm{mg\,m^{-3}}$, a threshold use by Zhai et al. (2011) to define phytoplankton bloom. The node #5 that occurred consistently through the winter months and had the highest assignment in the time series (20.1%), was used to delineate the winter chl-a plume region used in further analysis

(thereafter referred to as SOM5, Figure 3.

## 2.5 Grey Seal, Nitrogen and chlorophyll-*a* standing stocks: seasonal and interannual variations

### 2.5.1 Decadal change in grey seal abundance and a conceptual model of seasonal variation in presence

Sable Island is the largest grey seal breeding colony in the world. Outside of breeding season the island is an important haul out for seals that forage on the eastern Scotian Shelf. The breeding season starts in early December and lasts until February. The

population of grey seals peaks on Sable peaks in late January, with the accumulation of newly weaned pups, lactating females and males. While there are no counts of the entire seal population during the breeding period, the number of pups produced has been monitored since the 1960s with the last pup production estimate collected in 2016. At that time the pup production had increased from 30,000 in 1998 to roughly 90,000 in 2016 (den Heyer et al., 2020). We have developed a conceptual model to describe the seasonal variation in the number of adult seals (i.e., age one and above) hauling out on the island based on

the estimates of the number of pups and age 1+ animals from the assessment population model of Hammill et al. (2017) and the analysis of satellite-telemetry data from more than 100 seals that were tagged on Sable Island (O'Boyle and Sinclair, 2012; Breed et al., 2013). The seal population size for 2017 and 2018 was estimated by assuming a linear increase of 5% from previous years, and the seal population was then averaged over 5-year periods to remain consistent with the satellite observations for comparison. Roughly 80% of the seals tagged on Sable Island used the eastern Scotian Shelf as foraging

habitat (O'Boyle and Sinclair, 2012), and outside the breeding season seals spend about 20% of their time hauled out (Breed





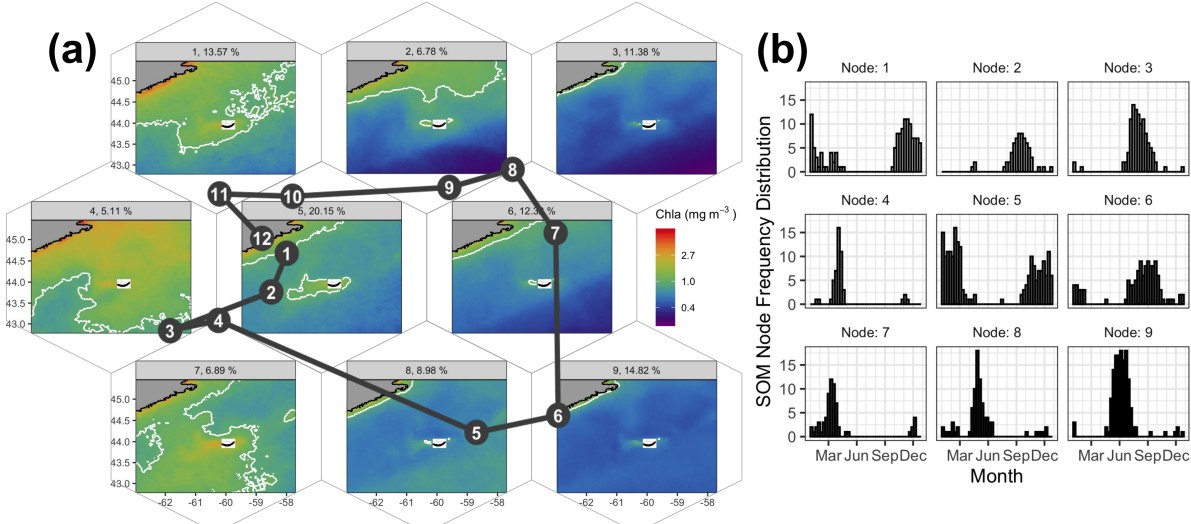

**Figure 3.** Results of the SOM, shown hexagonally to illustrate the spatial structure of the SOM layout (a). The node number and percent occurrence in the time series are shown with the accompanying patterns of chl-*a*. The white contour delineates 1.0 mg m$^{-3}$. The centroid of each month in SOM space is shown as the dark grey path, starting with January (1) near the center, proceeding counter-clockwise to December (12). The number of images per node and per week are shown as histograms on the right (b).

et al., 2013). Breed et al. (2013) estimated that in January males spend on average 80-85% of time hauled out and females about 40%. Our conceptual model does not differentiate between males and females the same, so we have estimated peak haul out as the average of time hauled out for males and females. Notably the timing of the peak haul out from satellite telemetry on the island in mid-January is consistent with birth distribution models (Bowen et al., 2007; den Heyer et al., 2020), estimates of mean pup birthdate from individually marked adults (Bowen et al., 2020), and the reproductive biology of the seals. The model is built around three hypotheses:

1. Seals start to arrive on SI in early December, steadily increasing until the end of January, after which seals rapidly leave the island,

2. Variation in seal annual abundance was modeled using a skewed log-normal distribution (Equation 1), with a peak abundance of 60% of the age 1+ animals present on SI in January (Figure A1). Note that, for practical purposes, the initiation day of the model is early October (i.e., October 8$^{\text{th}}$ in agreement with the 8-day composites),

3. For the remainder of the year, the seal population is constant at 16% of the age 1+ animals .

The model for the period P4 was therefore described as:

$$N_{seal}(d) = 36,930 + 195,000 \times \exp\left[-\frac{(\ln(d) - 2.3)^2}{1.1}\right], \tag{1}$$





where $d$ represents the day of year (varying from 365 to 1) and the number hauled out year round ($\sim$ 37,000) and the additional animals that haul out on the island during breeding (195,000) (Number of seals for period P1 to P3 are summarised in Table A1). The parameters 2.3 and 1.1 were defined to temporally align the arrival of the seals on SI, which starts to increase early December, and the peak occurring at the end of February (Figure A1). The seal population was then binned into 8-day means and averaged over 5-year periods to remain consistent with the satellite observations for comparison and the potential chl-$a$

standing stocks were computed for each period.

### 2.5.2 From seal population to chlorophyll-$a$ standing stocks, and comparison to satellite observations

We applied the nitrogen (N) excretion rate of 0.22 kg d$^{-1}$ to the seal abundance model on SI that Roman and McCarthy (2010) used for a study in the Gulf of Maine, which is also located on the eastern North American continental shelf close to Sable Island. This provided a means to convert the presence of seals on SI during winter in excretion of N in tonnes, which was further

converted into moles of N. Chlorophyll-$a$ concentration that can be sustained by N was computed using the conversion factor of 1.59 mg chl-$a$ (mmol N)$^{-1}$ from Matear (1995), which is in agreement with the value of 1.6 from Yentsch and Vaccaro (1958). This conversion provides a rough estimate of the upper limit of phytoplankton biomass that seal N-excretion can sustain in the surroundings of SI, assuming that all N would reach the water. Some of the mechanisms that describe the release of N in nature are not accounted for in the model, such as the amount of N that would be sequestered in the soil and used for the fauna, N that

will be leaked to the surrounding water through sand permeability (,which varies from a few centimeters to a few meters per hour, Lunne et al., 1997). We assumed that seal excretion is washed out to sea by atmospheric precipitation, which remained fairly constant over the annual cycle (Figure B1), noting that atmospheric transport of N as NH3 from seal colonies has been measured in other regions (Theobald et al., 2006). The release of N would also decrease when temperatures are negative as the soil is frozen and precipitation occurs as snow, which could introduce an additional lag between seal presence and N release

to the surrounding waters. The winter 8-day release of N was integrated over the entire breeding season (i.e., winter from December 1 to February 28) for all four periods (i.e., P1 to P4) to quantify the possible increase in chl-$a$ standing stocks.

Estimation of standing stocks of chl-$a$ from satellite observations was carried out for the SOM5 region, assuming that this region would be the most impacted by N-release from seal excretion. Chlorophyll-$a$ concentration (mg m$^{-3}$) was multiplied by the SOM5 surface area (m$^2$) and integrated over a depth of up to 50 m to provide standing stocks of chl-$a$ in tonnes. We

assumed an homogeneous chl-$a$ profile over the first 50 m, which is consistent with the strong mixing that occurs in this area in winter. Bi-weekly CTD profiles collected at a monitoring station located on the Scotian Shelf (i.e., HL2) over the last 20 years show that the mixed-layer depth reaches between 50 and 60 m in winter (Casault et al., 2020). The period 1999-2003 (i.e., P1) was chosen as a reference, as it provided the chl-$a$ standing stocks in the plume when the seal population was at its lowest number over the 1999-2018 period. We assume that the phenology of chl-$a$ during this early period was mainly due

to the oceanographic/environmental conditions (e.g., nutrient availability in the water column, temperature, light availability); however, it also accounts for the possible impact of seal N-release during the breeding season when the seal population was smaller, at about 100,000 individuals. The increase in 8-day chl-$a$ standing stocks not due to the oceanographic/environmental conditions for the period 2014-2018 (P4), assumed instead to be sustained by seal N excretion, was obtained by subtracting





the biomass between the periods P1 and P4. Note that the 8-day time series for both periods were smoothed (R smooth.spline
function) before subtraction to more easily examine the signal. Negative values of chl-*a* therefore indicated a net loss of biomass
standing stocks between the two periods of interest while positive values indicated a net gain of biomass. This provided a means
to compare the eight-day averaged chlorophyll-*a* standing stocks derived from satellite observations and model simulations for
the period P4.

## 3  Results and Discussion

### 3.1  Evidence of enhanced chlorophyll-*a* concentration around Sable Island

The Scotian Shelf is a productive environment with an annual mean chl-*a* of about 0.75 mg m$^{-3}$ ($\pm$0.38 mg m$^{-3}$) observed
by ocean colour satellites. Phytoplankton biomass is subject to a strong seasonal cycle with a spring bloom representing a
major food input into the ecosystem, and a secondary fall bloom triggered by the replenishment of nutrients to the surface-
lit layer as a result of physical forcing (Song et al., 2010). While this general progression is well established, the mesoscale
patterns of phytoplankton dynamics are more complex given the complicated bathymetry and hydrodynamics of the Scotian
Shelf. For instance, the effect of the numerous banks and channels on the hydrodynamics result in spatially distinct timing and
intensities of the spring bloom throughout the surface waters of the shelf (Casault et al., 2020) and an overall patchiness in
phytoplankton growth and decay (Denman and Platt, 1976). However, a feature that remains constant is the moderate to high
surface phytoplankton biomass that occurs leeward of Sable Island in comparison to its surroundings as observed by satellite
OCRS (Figure 1). While this has been documented in previous studies (King et al., 2016; Zhai et al., 2011), the temporal and
spatial dynamics of this plume off Sable Island have not been examined in details using a 21-year time series of the satellite
record.

The three large boxes delineated on the shelf show a clear south to north gradient in terms of chlorophyll-*a* concentration and
timing of the spring bloom (Figure 2a). The SW box shows the lowest annual mean of chl-*a* (0.67 $\pm$ 0.03 mg m$^{-3}$) compared
to the SI (annual mean of 0.87 $\pm$ 0.17 mg m$^{-3}$) and NE (annual mean of 0.87 $\pm$ 0.17 mg m$^{-3}$) boxes and the earliest spring
bloom (week 10, 11 and 12 for the SW, SI and NE boxes respectively) and the lowest spring bloom magnitude. In contrast, the
NE box exhibits the highest chl-*a* and the latest bloom initiation, in agreement with the northward progression of the spring
bloom (Siegel et al., 2002) even at such a small latitudinal scale (i.e, 2°N). The higher magnitude of the bloom in the NE box
relative to the SI and SW boxes might be explained by local processes, such as the presence of high winter nitrate, as shown in
Zhai et al. (2011).

The absorption coefficient follows a similar pattern to chl-*a* (Spearman correlation of 0.93, 0.88 and 0.88 for NE, SI and SW
respectively with p-value lower than 0.01, Figure 2b), suggesting that, away from terrigenous inputs, detritus and dissolved
organic matter originate from phytoplankton degradation, consistent with the definition of case I waters (Morel and Prieur,
1977). The backscattering coefficient at 443 nm shows a very different pattern than chl-*a* and $a_{dg}(443)$ with values remaining
relatively high in winter and early spring (weeks 1 through 13) followed by a small increase corresponding to the peak of the
spring bloom and a decrease in late spring at the end of the spring bloom (Figure 2c). The backscattering coefficient peaks in the



summer between week 24 and 26 (depending on the box) and continuously decreases thereafter until late fall. Backscattering magnitude is related to abundance and inversely related to particle size (Slade and Boss, 2015), and here the timing of maximum backscattering is consistent with the time of year when large phytoplankton such as diatoms and dinoflagellates reach their
minimum abundance, while small flagellates are most abundant. The mean $b_{bp}(443)$ in the SI box is higher than the ones in SW and NE during the winter, which is consistent with physical forcing that would trigger resuspension of sand and mineral particles around SI. When examining the 12 small boxes to the leeward side of Sable Island (Figure 2, colored lines), the seasonal cycles of chl-$a$, $a_{dg}(443)$ and $b_{bp}(443)$ tell a different story. The location of these boxes were chosen to emphasize the higher phytoplankton biomass southwest of the SI than elsewhere without a priori knowledge of the plume spatial distribution).
Boxes 7 to 12 show higher chlorophyll-$a$ concentration and $a_{dg}(443)$ than the control and SI boxes in winter and early spring (weeks 41 through 13), while boxes 1 to 6 show values similar to the control boxes except boxes 2 and 3 in weeks 1 to 10, which exhibit values slightly higher than the control boxes. In particular, boxes 10, 11 and 12 show the highest chl-$a$ through the winter, with a sharp increase that starts around week 4, followed by a slight decrease until week 6 when chl-$a$ increases until week 11, that is not evident in the other boxes. For the remainder of the year, chl-$a$ and $a_{dg}(443)$ in all 12 boxes follow
the same pattern as the control boxes SW and NE, without obvious enhanced chl-$a$.

The main impact of the island in the enhancement of chl-$a$ and $a_{dg}(443)$ happens from late fall to early spring in the leeward region of SI. The location and extent of the plume is consistent with the hydrodynamics of the area as revealed by model simulation (Brickman and Drozdowski, 2012) and comparison to the global sea surface velocity dataset (OSCAR).The annual depth-averaged Labrador current follows the continental slope along a southwest axis at a speed of about $20\,\mathrm{cm\,s^{-1}}$
, which is heavily reduced when reaching Sable Island bank, flowing southward with a much lower speed of about a few $\mathrm{cm\,s^{-1}}$ ($< 5\,\mathrm{cm\,s^{-1}}$ in OSCAR dataset). Current-induced kinetic energy has a strong seasonal cycle that is highest in fall and winter on Sable Island bank, and lowest in summer, while the spring values lie in-between (Brickman and Drozdowski, 2012, p. 18-21). The Labrador Current flowing southwest adjacent to the island and parallel along the shelf break, increases beginning in October, peaking in February and March, and declines until June (Loder et al., 1997). As such, there is a strong
relationship between the chl-$a$ and $a_{dg}(443)$ plume spatial extent and the hydrodynamic circulation of the area, particularly when contrasting boxes 10 through 12, which have higher winter chl-$a$ than the other boxes on the Sable Island Bank. Previous studies have indicated that shoaling on the relatively shallow Sable Island Bank provides nutrients to the water column and may explain the chl-$a$ enhancement (Zhai et al., 2011) observed here. However, if this were the case, we would expect greater similarity between all boxes on the bank. Furthermore, the high $b_{bp}(443)$ during the winter through early spring suggests that
the water masses in boxes 11 and 12 originate from Sable Island. Indeed in the presence of high concentrations of particles, OCRS estimations of IOPs are confounded, leading to greater uncertainty in the corresponding $a_{dg}(443)$ estimates. This may be the reason the peak in chl-$a$ during week 4 is not reflected in the $a_{dg}(443)$ for boxes 11 and 12, but is present for box 10. However, in terms of supporting the IME, the greater $b_{bp}(443)$ signal demonstrates the impact of water flow direction, even if $a_{dg}(443)$ may be underestimated.





## 3.2 Timing of biomass peak and seasonal trends

Details on the spatio-temporal dynamics of phytoplankton biomass on the Scotian Shelf, and in particular around Sable Island, are revealed when applying an unsupervised classifier to the entire time series of 8-day composite of chl-*a*. Self-organizing maps (SOMs) are a useful tool for extracting patterns in time series of satellite imagery and their frequency of occurrence (Richardson et al., 2003), with previous applications including pattern analysis of sea-surface height (Hardman-Mountford et al., 2003), spatial-temporal scatterometer and SST phenology (Richardson et al., 2003), and improved pixel classifications of satellite chl-*a* time series (Ainsworth, 1999). Here we applied the SOMs method to the satellite chl-*a* estimates to summarize the spatio-temporal dynamics of phytoplankton biomass in nine main patterns on the Scotian Shelf.

A common feature to all maps was a region of enhanced chl-*a* around SI that is more pronounced to the south-west of the island, in agreement with the climatology study (section 3.1). The largest extent of the plume occurred mainly in winter (Figure 3a, SOM node #5) with a mean surface area of chl-*a* > 1 mg m$^{-3}$ of $4.10^3$ km$^{-2}$. The threshold of $1\,\mathrm{mg\,m^{-3}}$ was selected arbitrarily as an indicator of high chl-*a* event based on the climatology of the large boxes (Figure 2) and for consistency with other studies in this area (e.g., Zhai et al., 2011). This chl-*a* plume around SI persists in the spring, even during the spring bloom that develops over most of the shelf (SOMs # 4 and 7), and until the spring bloom terminates (SOMs # 8 and 9). In summer, from June to August, a low background of chl-*a* on the Scotian Shelf emphasizes the enhanced chl-*a* to the southwest extent of SI, somewhat with a smaller surface area than in winter (SOMs # 6 and 3). Finally, from September to November (SOMs #1 & 2), the chl-*a* remains distinctively higher than its surroundings despite an overall increase in biomass (i.e., fall bloom). The node #5 is the most frequent, occurring 20% of the time in the entire time series (Figure 3b). The SOM analysis strongly supports the hypothesis of the IME impact on chl-*a* without examining underlying mechanisms. The extent of the plume and the chl-*a* within its boundaries follow a seasonal cycle in agreement with the current seasonal cycle (see section 3.1) and remains independent of the phytoplankton phenology on the rest of the Scotian Shelf as revealed by the two control boxes.

**Table 1.** Slope and p-values of the linear regression of annual chl-*a* (mg m$^{-3}$ y$^{-1}$), $a_{dg}(443)$ (m$^{-1}$ y$^{-1}$) and $b_{bp}(443)$ (m$^{-1}$ y$^{-1}$) versus time in years for each region.

| Region | chl-*a* slope$\times 10^{-5}$ | p-value | $a_{dg}(443)$ slope$\times 10^{-6}$ | p-value | $b_{bp}(443)$ slope$\times 10^{-7}$ | p-value |
|---|---|---|---|---|---|---|
| NE | 1.61 | 0.05 | 4.6 | < 0.01 | -0.91 | < 0.01 |
| SOM5 | 1.32 | 0.10 | 3.4 | < 0.01 | -0.1 | < 0.01 |
| SW | 0.682 | 0.15 | 2.4 | < 0.01 | -0.65 | < 0.01 |

The annual seasonal trends for the 20-year time series were calculated for the SW and NE and SOM5. Interestingly, chl-*a* did not exhibit any significant trends in both control boxes and SOM5 (Table 1). On the other hand, $a_{dg}(443)$ shows a positive significant trend in all three regions of interest (ROIs) with slopes of the same order of magnitude (Table 1), while $b_{bp}(443)$ showed negative trends in all three regions. The negative trend around SI (i.e., SOM5) is an order of magnitude smaller than





for the NE and SW boxes. The significant positive trends observed for $a_{dg}(443)$ while chl-$a$ does not show a trend, suggest that the satellite-derived chl-$a$ signal is not contaminated by $a_{dg}(443)$, or both properties would have shown a positive significant trend. In a similar manner, the decrease in $b_{bp}(443)$ suggests that long term variation in biomass (i.e., chl-$a$) are not the result of particle resuspension around SI. The increase in $a_{dg}(443)$ in the vicinity of SI could be partially explained by the release of feces by Seal in the water.

**Table 2.** Slope and p-values of the linear regression of annual chl-$a$ (mg m$^{-3}$ y$^{-1}$), $a_{dg}(443)$ (m$^{-1}$ y$^{-1}$) and $b_{bp}(443)$ (m$^{-1}$ y$^{-1}$) versus time in year for each season and region.

| Region | chl-$a$ slope | chl-$a$ p-value | $a_{dg}(443)$ slope | $a_{dg}(443)$ p-value | $b_{bp}(443)$ slope $\times 10^{-5}$ | $b_{bp}(443)$ p-value |
|---|---|---|---|---|---|---|
| | | | **Spring** | | | |
| NE | 0.014 | 0.22 | 0.0026 | 0.02 | -2.0 | 0.16 |
| SOM5 | 0.004 | 0.62 | 0.001 | 0.12 | -5.0 | < 0.01 |
| SW | 0.002 | 0.65 | 0.001 | < 0.01 | -3.0 | < 0.01 |
| | | | **Summer** | | | |
| NE | 0.003 | 0.40 | 0.0007 | < 0.01 | -5.0 | 0.02 |
| SOM5 | 0.001 | 0.76 | 0.0008 | < 0.01 | -2.0 | 0.20 |
| SW | 0.001 | 0.55 | 0.0007 | < 0.01 | -1.0 | 0.35 |
| | | | **Fall** | | | |
| NE | 0.0007 | 0.91 | 0.0014 | < 0.01 | -5.0 | < 0.01 |
| SOM5 | 0.0042 | 0.48 | 0.001 | < 0.01 | -7.0 | < 0.01 |
| SW | 0.0019 | 0.66 | 0.0007 | 0.03 | -3.0 | < 0.01 |
| | | | **Winter** | | | |
| NE | 0.015 | < 0.01 | 0.0025 | < 0.01 | -3.0 | < 0.01 |
| SOM5 | 0.028 | < 0.01 | 0.0028 | < 0.01 | -1.0 | 0.60 |
| SW | 0.0114 | < 0.01 | 0.0016 | < 0.01 | -2.0 | 0.04 |

When partitioning each year into seasons, chl-$a$ trends become significant in the winter for all three ROIs (Figure 4a and Table 2), while no trends in chl-$a$ are observed in these ROIs for the other seasons. It is noteworthy that the rate of increase of chl-$a$ over the period of observation for the plume near SI (i.e., SOM5) is twice that of the control boxes. The $a_{dg}(443)$ linear trend is also positive and significant in all three ROIs in winter (Figure 4b). The highest increase was observed in winter around SI (slope of 0.0028 m$^{-1}$ y$^{-1}$) as for the chl-$a$ trend; the second highest positive linear trend is also observed in winter in the

NE box (0.0025 m$^{-1}$ y$^{-1}$). For the other seasons, the increase, when significant, was much smaller. In summer, the slope is significant and around 0.0007 m$^{-1}$ y$^{-1}$ for all three ROIs, while for spring and fall the results are more variable with only the SW region showing a significant positive trend in summer, and the SOM5 and NE regions showing a significant positive trend

in fall. As for the annual trend, the backscattering coefficient also shows significant declines in all seasons and ROIs except

for the NE in spring, SI and SW in summer and SI in winter (Figure 4c). The fact that the particulate backscattering in SOM5

does not show any significant trend in winter suggests that the increase in chl-*a* observed by satellite is not due to particle

resuspension, but rather by new production.

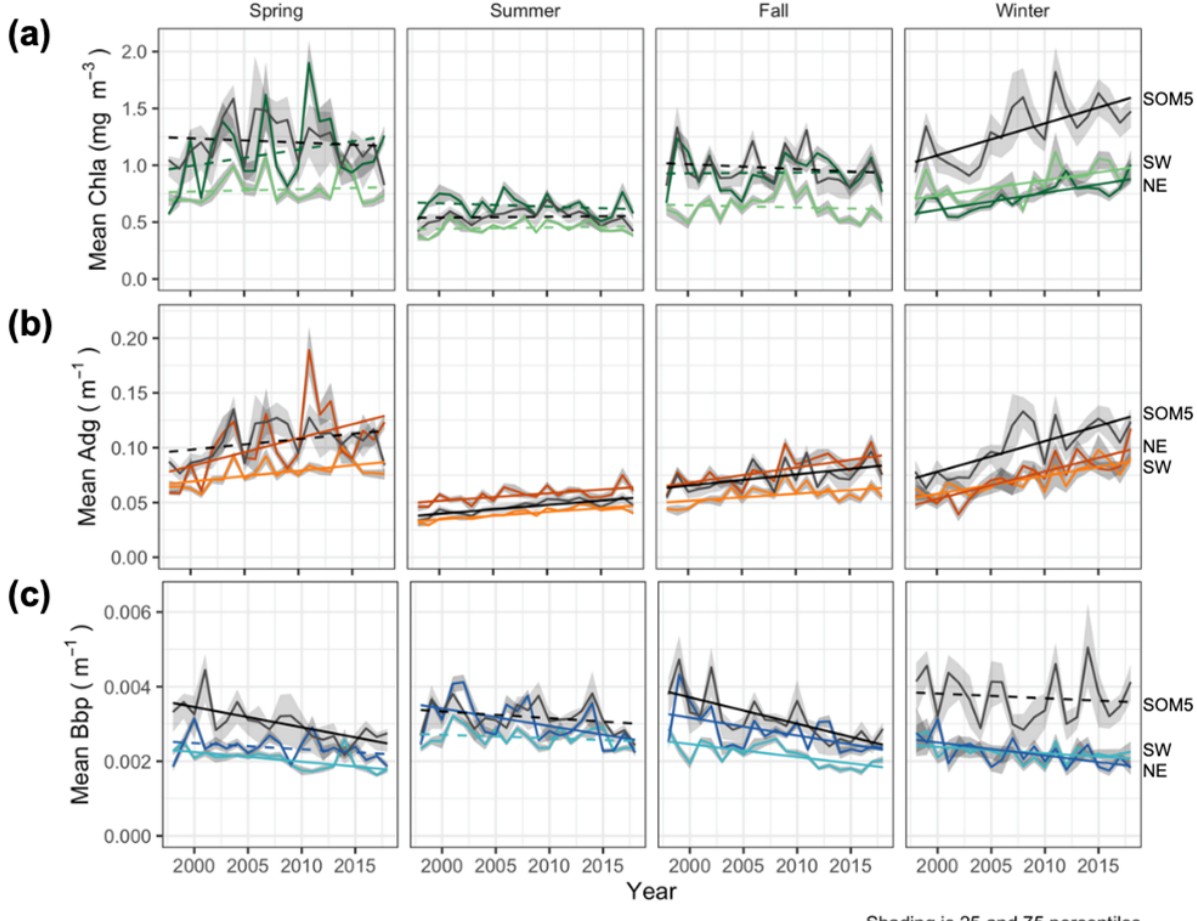

**Figure 4.** Seasonal mean chl-*a* (a), $a_{dg}(443)$ (b) and $b_{bp}(443)$ (c) for the plume defined by SOM (SOM5) and control boxes (NE and SW). Statistically significant trends (p < 0.05) are shown as solid lines, while non-significant are dashed. Line regions labelled on the right.

When averaging the winter chl-*a* over a 5-year period, and examining the satellite-based chl-*a* as biomass standing stocks

over the four periods (Figure 5), the increase in biomass standing stocks is even more striking, as it reveals both the increase in

phytoplankton biomass within the plume and also the increase in the plume surface area. Our study shows that not only does

Sable Island have a local effect on the surrounding ecosystem as enhanced chl-*a* is observed in the leeward region of the island,



but also that the phytoplankton biomass has increased in winter over the last two decades at a rate not matched by the other regions (i.e., control boxes).

### 3.2.1 Simultaneous increase in seal abundance and phytoplankton biomass, a coincidence?

Satellite observations of chlorophyll-*a* concentration and other bio-optical properties provide a unique opportunity to link
phytoplankton biomass dynamics around Sable Island to the seasonal and decadal variations in the seal population. While other effects surely contribute to the enhanced chl-*a* around SI, we have tried to eliminate some of the mechanisms by comparing the SI environment to two control boxes and by analysing the spatio-temporal variation of other bio-optical properties (i.e., $a_{dg}(443)$ and $b_{bp}(443)$) that are proxies for hydrographic mechanisms. Possible hypotheses to explain this phenomenon are 1) the increase is due to particle resuspension (e.g., sand), 2) terrestrial input of nutrients or degradation of vegetative matter, or
3) shoaling on Sable Island Bank and upwelling along the continental shelf break. However, none of these mechanisms support such a strong and local increase of chl-*a*, as not only the mean chl-*a* has increased within the plume, but the surface area of the plume has been multiplied by a factor five over the period of observations (Figure 5). The resuspension of mineral particles (hypothesis 1) in the water column does not support this increase as we have demonstrated that the particulate backscattering coefficient, an indicator of mineral particle abundance in shallow water, has decreased or has remained stable around SI, as in
the other control boxes. In particular, during the winter, when chl-*a* has increased in the SI plume twice as fast as in the control boxes, $b_{bp}(443)$ did not exhibit any trends. Sable Island itself is partially vegetated mostly with marram grass, and does not have significant overland flow due to the high infiltration rate of the sand, which discards hypothesis 2, which would assume an increase in runoff from the island. The position of SI near the continental shelf may explain some of the elevated chl-*a* (hypothesis 3), with slope waters having greater nutrients from horizontal and vertical mixing Zhai et al. (2011); however, if
that was the case, this phenomenon would occur along the entire slope and also in the SW box, which was not evidenced. Shoaling on Sable Island Bank has been documented in Zhai et al. (2011) and postulated as a mechanism for higher chl-*a* than in the surrounding areas, but this does not explain the increase over time. In addition, repeated measurements of nutrients over the entire Scotian Shelf over the last 20 years have revealed a continuous decrease in nitrate concentration since about 2012 in agreement with an increase in stratification (Casault et al., 2020).

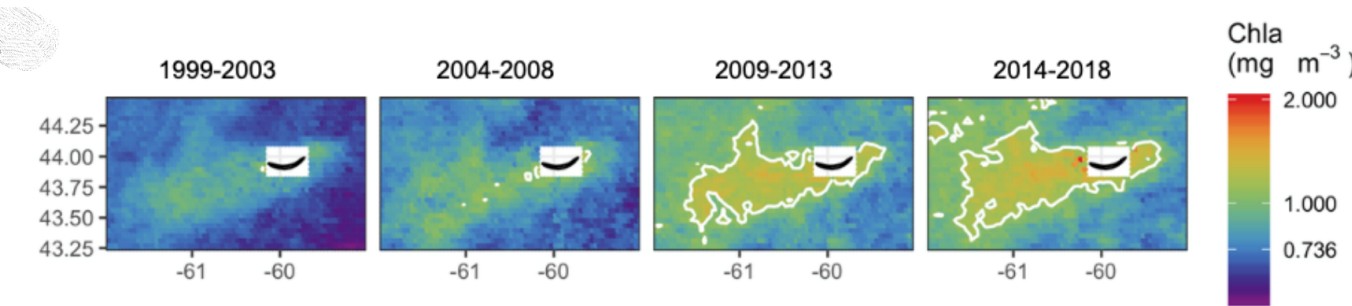

**Figure 5.** Mean chl-*a* in winter for the four periods of interest. The white solid line represent the $1\ \mathrm{mg\ m^{-3}}$ contour.





While the hypotheses stated above do not provide a satisfactory explanation of the spatio-temporal pattern occurring leeward of SI, an interesting parallel to the satellite observations over the same time period is the large increase in the population of grey seal (*Halichoerus grypus*), particularly during the breeding season in the winter months (December through February, Figure 6). Over the 20-year period of observation, Sable Island has seen its population of seals increase during the breeding season by 300% from about 100,000 in the late 90s to more than 300,000 in recent years (Hammill et al., 2017). Pup production over

this period has been increasing at about 5-7% per year (den Heyer et al., 2020). Fecal matter from marine mammals including pinnipeds and whales liberates nitrogenous compounds including $NH4^+$ (Roman and McCarthy, 2010) to the environment. Fertilization of the ocean by marine mammals and birds has been demonstrated in several regions of the world (e.g. Roman and McCarthy, 2010; Laver et al., 2012; Mccauley et al., 2012; Wing et al., 2014). In the Gulf of Maine, adjacent to the Scotian Shelf, Roman and McCarthy (2010) estimated that whales and seals provide up to $2.3 \times 10^4$ metric tonnes of N per

year to the surface waters, citing a rate of 0.22 kg N per day excreted by grey seal individuals. Further, NH3+ fertilization of a coastal ecosystem near a large pinniped colony through atmospheric transport and deposition has been documented elsewhere (Theobald et al., 2006). A parallel was therefore made between the contribution of the growing population of grey seals on SI and the increase in phytoplankton biomass around Sable Island, particularly considering that $NH4^+$ has been shown to be used preferentially to Nitrate by phytoplankton on the Scotian Shelf (Cochlan, 1986). Previous research has also provided evidence

that the seal breeding colony is fertilizing the vegetation on the island and that this helps to support the horse population (McLoughlin et al., 2016).

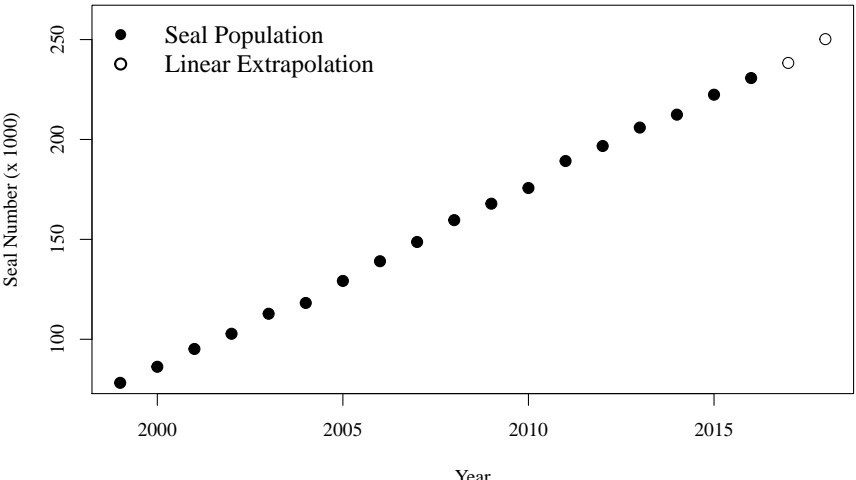

**Figure 6.** Number of seals on SI in winter as a function of time in years.





### 3.2.2 Seal abundance on Sable Island and chlorophyll-*a* standing stocks in SOM5

The difference in chlorophyll-*a* standing stocks between the period 1993-2003 and the period 2014-2018 is most important from winter to early spring (Figure 7) with the most recent period (P4) higher than the reference period (P1). The maximum
difference is reached early March with an excess of 133 tonnes of chl-*a*. From late spring to fall, the difference between the two periods ranges between $-50$ to $+10$ tonnes of chl-*a*. The normalised chl-*a* phenology in the NE box shows low relative chlorophyll-*a* concentration in winter, as it is expected on the Scotian Shelf, which contrasts with the high biomass and chl-*a* that occurs in the SOM5 region. The results observed in the SOM5 region are in agreement with the decadal trend analysis that did not show any trends in chl-*a* in spring, summer and fall in the SOM5 region while winter showed the highest rate of
increase ($0.028\,\mathrm{mg\,m^{-3}}$). Regardless, the presence of seals on the island has been increasing during spring, summer and fall. The faster rate of increase in winter could suggest that there has been a change in the distribution of seals as the population has grown with a smaller proportion of seals using Sable Island as a haul out location outside of the breeding season. Here, by taking the difference of the satellite 8-day composite chl-*a* standing stocks averaged over 5-year period, we have an insight into the temporal increase in chl-*a* during the winter, which shows a continuous increase in chl-*a* standing stocks in SOM5 from
early December to the end of February, followed by a rapid decrease. Except for the sudden increase of chl-*a* and standing stocks from early December to late February, the chl-*a* phenology shows a similar pattern for P1 and P4 with a spring bloom that occurs in mid-March and lasts for about 2 months until mid-May, and a fall bloom that starts in late September. The spring bloom during P4 is somewhat masked by high chlorophyll-*a* concentration that starts to occur in early December. In summer, the standing stocks of chlorophyll-*a* are similar between both periods of observations.

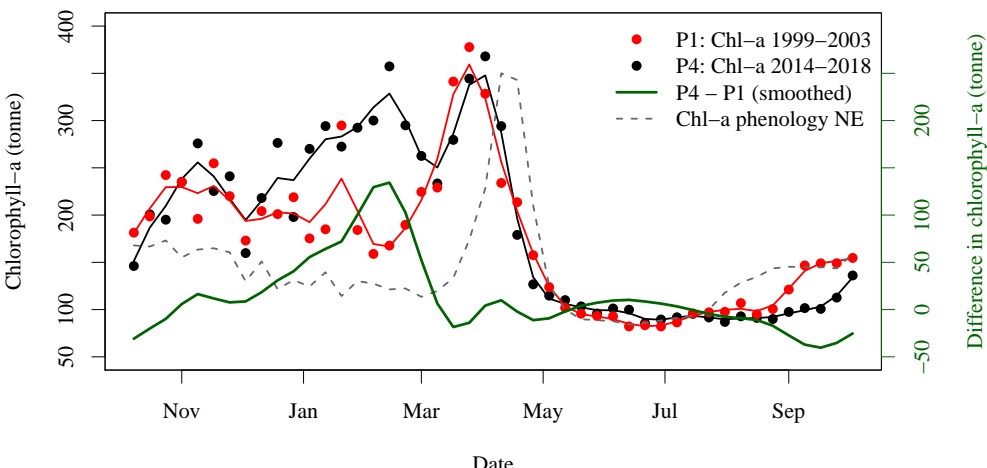

**Figure 7.** Eight-day composite images (solid circles) and smoothing (solid lines) of chlorophyll-*a* standing stocks in the SOM5 region for the periods 1999-2003 (red) and 2014-2018 (black). The green solid line corresponds to the difference between the two periods. The grey dahsed line corresponds to the Chl-*a* phenology in the NE box, assumed free of N fertilisation by seals.





A simple model was developed to understand and quantify the possible impact of grey seals on the N budget and chlorophyll-
*a* standing stocks around SI during P4 (Figure 8). While the mathematical formulation of the model was rather arbitrary, we
constrained the dynamics of seal hauling on the island with current knowledge. Grey seal abundance year long on SI is about
16% of the total adult population, currently about 40,000 individuals. This number increases continuously from December
to reach its maximum in January where about 60% of the adult population is expected to be hauled out, which would be at

peak about 150,000. Seals defer digestion and therefore defecation during rest time, suggesting that they may, indeed, relieve
themselves on SI or its surroundings rather than during foraging in the ocean (Sparling et al., 2007). The daily pattern of N
release, which was further converted into chlorophyll-*a* standing stocks following Roman and McCarthy (2010) and Matear
(1995) is a multiplicative term based on an average rate per individual. As we expect that the larger male grey seals are spending
more time using Sable Island as a haul out, this rate is a conservative estimate of N flux per inidividual.. We found that for the

Period P4, annual nitrogen released by seals could support about 500 tonnes of chlorophyll-*a*, while the biomass standing stocks
sustained by seals during the breeding period equals 232 tonnes. About 46% of the potential chlorophyll-*a* standing stocks due
to seal fertilisation would occur during the breeding season. There is a good temporal agreement between the standing stock of
chlorophyll-*a* concentration derived by satellite and modeled with a Pearson correlation coefficient of 0.54, which increases to
0.86 when using an 24-day lag (i.e., corresponding to three 8-day period), with the chlorophyll-*a* concentration reaching its peak

later that the peak in N release. We have also estimated the winter chl-*a* standing stocks using both satellite-observed and the
model of hauled out seals in four 5-year blocks. The seal supported standing stock accounts for about half the increase between
P4 and P1 derived by satellite observations (i.e., 757 tonnes, Figure 8). Given the number of assumptions and parameters used
for both estimates, this agreement provides further evidence that seal-derived N could support the observed plume. Here we
used the estimated N excretion based estimates of daily consumption and metabolic models (Roman and McCarthy, 2010).

These were annual estimates that do not account for breeding behavior or sexual-dimorphism which could have an impact
on the estimates of N release at the breeding colony, where females fast during lactation and much larger males spend higher
proportion of time hauled out. Similarly, for the satellite-observed chl-*a*, we used conversion factors from the literature to derive
the daily nutrient release and chl-*a* standing stocks. We used a conversion factor of N to chl-*a* of $1.59 \, \mathrm{mg \, Chl}$-$a \, (\mathrm{mmol \, N})^{-1}$
in agreement with Yentsch and Vaccaro (1958); Matear (1995), however, Li et al. (2010) found a conversion factor that varies

from $1.1 \, \mathrm{mg \, Chl}$-$a \, (\mathrm{mmol \, N})^{-1}$ on an intra-daily scale to $1.4 \, \mathrm{mg \, Chl}$-$a \, (\mathrm{mmol \, N})^{-1}$ at a weekly scale during the spring bloom,
using this value would further reduce the chl-*a* standing stocks that could be sustained by seals. The satellite observations are
mainly constrained by the assumption of an homogeneous profile of chl-*a* from 0 to $50 \, \mathrm{m}$ and the delineation of the plume.
Here we decided on a $1 \, \mathrm{mg \, m}^{-3}$ threshold to delineate the plume in the node #5 of the self-organising maps, which is consistent
with the description of a phytoplankton bloom on the Scotian Shelf. Using a smaller threshold could drastically decrease the

chl-*a* standing stocks estimates; however the general patterns and notably the temporal variation would remain the same. To
test the mechanism *in situ* sampling to estimate the standing stock and the water chemistry surrounding SI is necessary. The
mathematical formulation describing seal haul out dynamics on the island captures the continuous increase and the expected
sudden drop in seal abundance on the island at the end of the breeding season. Both a linear and s-shape model formulation to
describe the winter seal haul out on SI followed by a sudden drop at the end of the breeding season were also tested (results not





shown here), and provided similar results. The lag observed between the peaks in seal abundance and phytoplankton biomass in the plume can be explained by several factors, including: 1) the transport of seal feces to the ocean, which will certainly takes a few days and depends on atmospheric forcing, soil permeability and seastate and tide, with storm surges inundating the beach and releasing organic material, 2) the negative air temperatures that occur between the end of January and February (Figure B2) would reduce the diffusion of N from SI to the adjacent waters and 3) day length, which increases by about an hour

from about 9h45min to 11h05min hours between the January and the end of February, which will increase primary production over this period as N is not a limiting factor. Undoubtedly, other environmental factors contribute to the production of biomass in the area of interest (e.g., atmospheric forcing, timing and strength of stratification, light availability), that are not taken into account in our conceptual model.

### 3.2.3 Synchronized decadal increase in seal population, extent and concentration of the chlorophyll-*a* plume around
465       **SI**

Seasonal analysis of chl-*a* standing stocks and grey seal population on SI has evidenced a causal link explained by N release into the environment by seals that sustains enhanced phytoplankton biomass during the late fall and winter. Another remarkable phenomena is the winter increase in chl-*a* standing stocks in the leeward region of SI (SOM5) that is consistent with the increase of seal population on SI over the last 20 years. From the late 1990's to the late 2010's the population has continuously increased

at a rate of about 5-7% per year (Figure 6 den Heyer et al., 2020). We demonstrated that the average chl-*a* increased within the plume southward of Sable Island at a faster rate than the surrounding control boxes in winter. In terms of the five-year average standing stocks of chl-*a* over the four periods of interest, P1 to P4 (Figure 9), a quasi-linear increase in chl-*a* standing stocks is observed, except for during the period P2, which slightly departs from linearity. The high values for P2 are due to a sudden increase in chl-*a* standing stocks during a large bloom occurring in early December, a phenomenon that may be

explained by natural variation in phytoplankton biomass, and notably a short growth period that can occur on the Scotian Shelf when conditions are favorable. The increase in chl-*a* standing stocks due to climate forcing was computed by taking the value during P1 and subtracting the chl-*a* standing stocks due to N-fertilisation by seal and then applying a mean 5-year rate of increase of $0.055 \, \text{mg m}^{-3} (5\text{-year})^{-1}$ taken from the annual rate of increase at the SW box (i.e., $0.01 \, \text{mg m}^{-3} \text{y}^{-1}$). This provided chl-*a* standing stocks of 2323, 2562, 2592 and 2680 tonnes for P1N, P2N, P3N and P4N respectively, which

corresponds to an increase of chl-*a* standing stocks of 128, 263 and 405 tonnes between each 5-year time period (Figure 9). During the same periods, the contribution by seal fertilisation to chl-*a* standing stocks was 104, 152, 205 and 254 for P1, P2, P3 and P4 respectively. The sum of the chl-*a* standing stocks due to the natural increase and seal fertilisation match the total chl-*a* standing stocks observed using satellite ocean colour remains within a few tonnes (Figure 9). The main difference occurs during the period P2 with a difference of 110 tonnes between both estimations. For the period P4, the combined climate forcing

and seal fertilisation provide a larger estimate than the one observed by satellite. The linear increase due to climate forcing might be too simplistic as Liu et al. (2018) showed that the increase in chl-*a* on the Scotian Shelf slowed down after 2012, but the consistency between both estimates remains remarkable. The contribution by seal N-fertilisation to the total chl-*a* standing stocks in SOM5 has increased from 4.3 to 8.7% between P1 to P4 and the contribution by climate forcing increased from 4.7





to 13.8% between P2 and P4. As for the seasonal cycle, the satellite estimates provided here depend on the delineation of the

plume southward of SI; the value selected here (i.e., $1\,\mathrm{mg\,m^{-3}}$) happened to provide a very good agreement between the model and the satellite observations. The correlation coefficient between the model- and satellite-derived chl-$a$ standing stocks is 0.89, such that while the agreement in magnitude is subject to the parameters in the model, the agreement in the temporal variation supports the contribution of seal N-fertilisation to the increase in observed standing stocks. This is another striking finding that advocates for the impact of seal fertilisation of the nearby ocean as no other mechanisms have supported the strong increase

of chl-$a$ in this region (see section 3.2). The mean seal population size has multiplied by 3 on average from approximately 100,000 in the late 1990's, to over 300,000 in the late 2016; during the same period, the standing stocks of chl-$a$ increased by 21%, which corresponds to an additional 506 tonnes of chl-$a$ in the ecosystem.

## 4    Conclusions

Sable Island, a large sand bar located at the edge of the continental shelf off eastern Canada, sees enhanced chlorophyll-$a$

and yellow substance concentration in its vicinity and in particular leeward of the island as revealed by satellite ocean colour. This finding is in agreement with the theory of island mass effect that has been demonstrated in other parts of the world. However, to our knowledge, it is the first time that it is observed in a mesotrophic environment as previous studies focused on oligotrophic waters of tropical and sub-Antarctic regions. The increase and phenology of phytoplankton biomass around SI was demonstrated by comparison with two control boxes located to the northeast (NE) and southwest (SW) of SI, away

from possible terrigenous inputs. We found that chlorophyll-$a$ concentration around SI showed a different annual cycle than the control boxes with higher values in late fall through winter. This was also the case for the absorption coefficient of yellow substances and the backscattering coefficient. Using an objective classification method (i.e., self organising maps), we were able to describe the spatio-temporal variation of chl-$a$ on the Scotian Shelf in general and around SI in particular. A distinct pattern of high chl-$a$ around SI, referred to as SOM5, was revealed by this analysis, and occurs from late fall through winter;

comparison with the chl-$a$ phenology in the control boxes showed that this pattern was different than the fall bloom and singular to SI. Trend analysis of satellite-derived chl-$a$, $a_{dg}(443)$ and $b_{bp}(443)$ over the last 21 years showed that, at the annual level, $a_{dg}(443)$ had increased at a similar rate in all three regions (i.e., SOM5, SW and NE), while $b_{bp}(443)$ decreased in all three regions, though at a much slower rate around SI. The trend was not conclusive for chl-$a$. When dividing the years and examining by season, mean chl-$a$ showed a significant increase in these three regions of interest in winter, but with a trend

twice as high for SI than for the SW and NE boxes i.e., $0.028\,\mathrm{mg\,y^{-1}}$.

     The location of SI (away from continental inputs) and the analysis of $a_{dg}(443)$ and $b_{bp}(443)$ seasonal cycles and decadal trends did not help explain the processes that led to the increase in chl-$a$ around SI, as no trends in particulate backscattering and DOM absorption were markedly different from the surrounding environment. The only drastic change that has occurred on SI is a significant, and notable, increase in the seal population that breeds on Sable Island from about 100,000 to more than

300,000 thousand individuals from 1999 to 2018. We developed a simple model that describes the seal occurrence on SI, in particular during the reproductive season (i.e., early December to end of February), and related the daily theoretical N release


due to seal excretion to chl-*a* standing stocks. Comparison of the model results with satellite-derived chl-*a* standing stocks was compelling, not only in terms of the seasonal cycle, but also to the decadal trends which showed an agreement with the seal population growth on SI and the increase in chl-*a* standing stocks nearby. In fact, not only the mean chl-*a* increased within the

plume located leeward of the island, but the plume surface area increased by a factor of five over the period of observation.

Our findings demonstrate that the island mass effect is not restricted to oligotrophic, often tropical, environments and that it can be significant in mesotrophic shelf waters. In addition, the continuous growth of the total seal population size on SI during the breeding season that has exceeded 300,000 individuals in recent years, and the associated N release in the surrounding waters can support chl-*a* standing stocks of 254 tonnes in winter, in agreement with standing stocks observed in the SI chl-*a*

plume in recent years. While we acknowledge the lack of *in situ* measurements in the surrounding waters of SI, the strong agreement between our model results and satellite observations support the important role that grey seals can play in the fertilisation of the ocean. Our results are consistent with Roman and McCarthy (2010), who found that marine mammals (both seals and cetaceans) could sustain $2.3 \times 10^4$ tonnes of N in the Gulf of Maine. The high concentration of seals on a small island for a short period of time represents ideal conditions to witness their impact on the marine ecosystem; however, this

impact is certainly diminished when they leave the breeding colony to forage over very large areas (i.e., the Scotian Shelf, Gulf of St. Lawrence, Gulf of Maine and Grand Banks). Our study shows the top-down control of a species of marine mammals on their surroundings and how it enhances the productivity of a small region, with implications for marine conservation and biodiversity.

*Author contributions.*

E.D. and C.d.N. initially discussed the idea of seal fertilisation around SI, E.D. designed the study. A. H. carried out all computations and statistical analysis related to satellite data. E.D. and C.d.N. developed the seal distribution model and chl-*a* standing stock budget. A.H and E.D. contributed equally to the drafting of the manusucript. A.H, C.d.H and E.D. all contributed to the editing of the manuscirpt.

*Competing interests.* The authors declare that there is no conflict of interest.

*Acknowledgements.* The study was supported by Fisheries and Oceans Canada as well as the Canadian Space Agency Climate Change Impact and Ecosystem Resilience program.





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

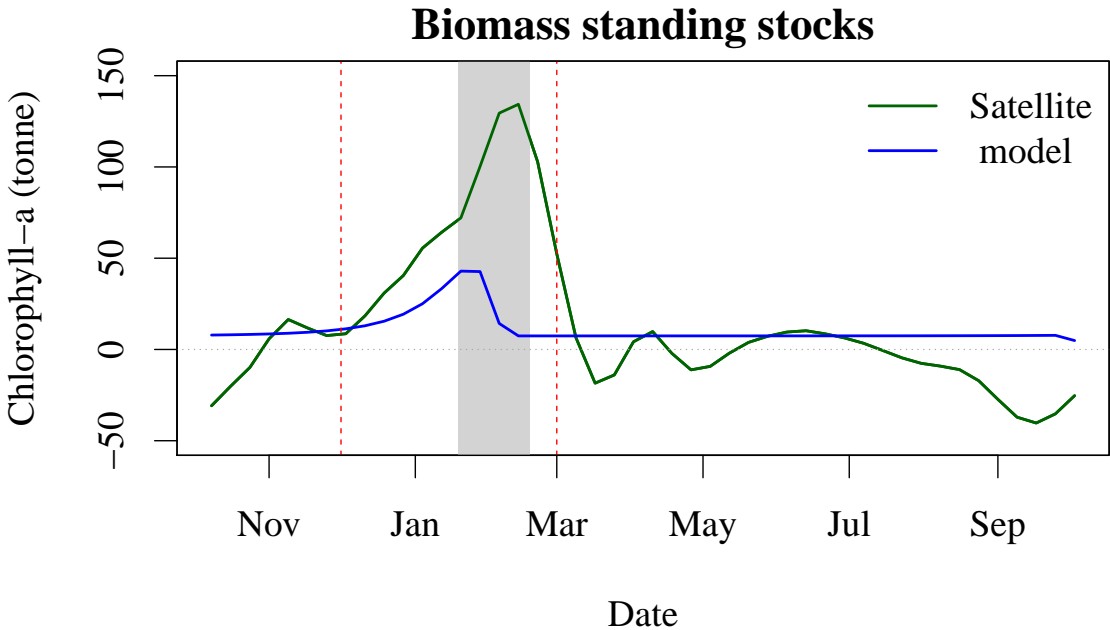

**Figure 8.** Increase in chl-*a* standing stocks between P1 and P4 (green solid line) and modelled chl-*a* standing stocks due to seal N-release in the environment (blue solid line). The vertical red dashed line correspond to the arrival and departure of seal on SI (i.e., breeding seasaon) and the light grey rectangle corresponds to the climatology of air temperature that is below zero.



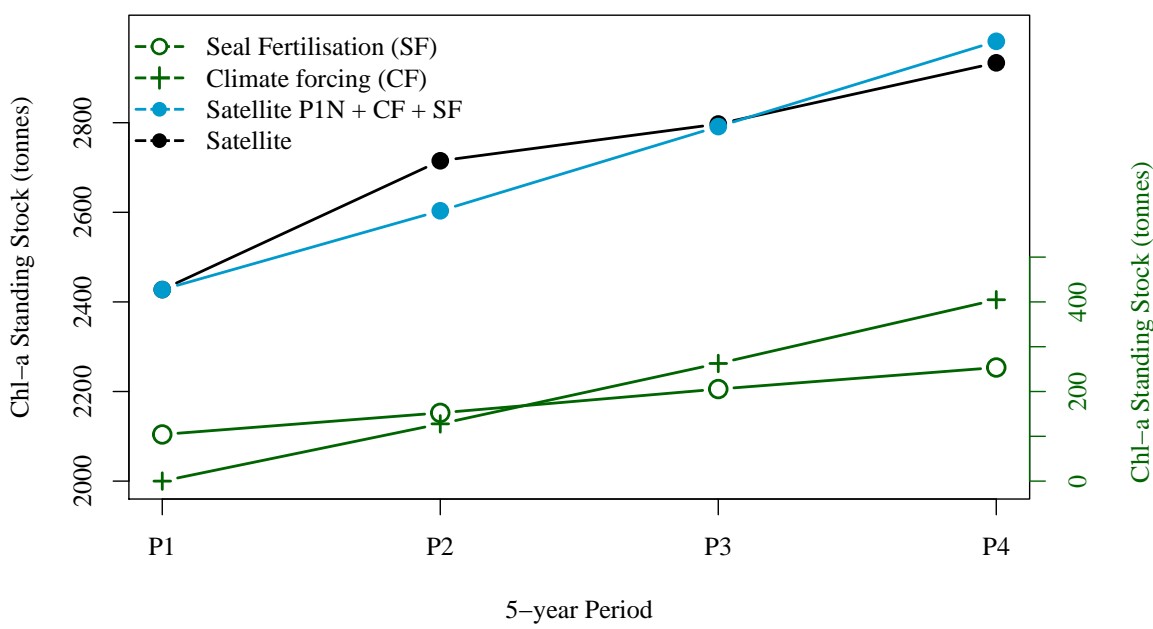

**Figure 9.** Five-year winter average standing stock of chl-*a* derived from satellite observations (solid black circle), from satellite observation for P1 extrapolated with climate forcing and seal N-fertilisation (blue solide circles), from climage forcing (green crosses) and from seal N-fertilisaton (green open circles) for the four periods of interest (black solid circles) and five-year average seal pup production (green solid circles).





## Appendix A: Seal haul out model

The annual seal population distribution on Sable Island was estimated using information on pup production from Hammill et al. (2017) and den Heyer et al. (2020). The mathematical formulation is described in section 2.5.1, equation 1. The model was also applied to the average seal adundance on SI for the periods P1 to P3 (Table A1) and used to generate the chl-*a* standing stocks

in Figure 9. Note that fluctuations around the mean seal abundance outside the breeding season is expected but not accounted for in the model.

**Table A1.** Seal haul out on SI all year long and during the breeding season use in the seal population dynamic model.

| Period | Seal Number | |
| :---: | :---: | :---: |
|  | Year around | breeding season |
| P1 | 15,204 | 80,000 |
| P2 | 22,232 | 117,000 |
| P3 | 29,935 | 158,000 |
| P4 | 36,931 | 195,000 |

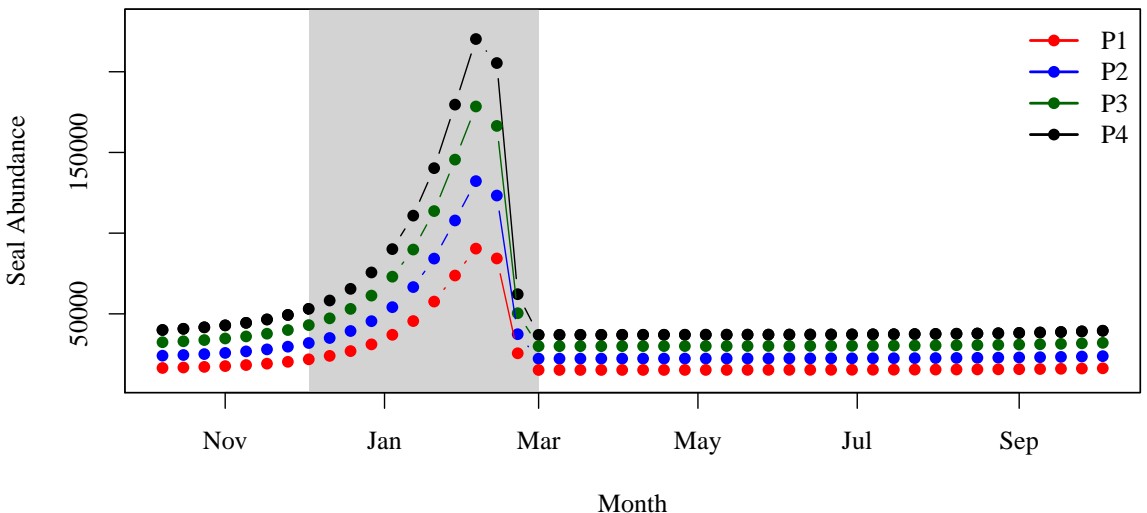

**Figure A1.** Annual cycle of seal hauls on SI for the four period of interest. The grey shaded area corresponds to the breeding season.



**Appendix B: Eight-day climatology of precipitation and temperature on SI.**

Daily data at the SI weather station were downloaded from the ECCC website
(https://climate.weather.gc.ca/historical_data/search_historic_data_e.html). Data from the station 8204700 (1998-2017) and
8204708 (2018) were used. The temperature and precipitation 8-day climatology was computed by averaging all data available
in a given 8-day period for all years. The standard deviation is indicated with the mean in both figures B1 and B2.




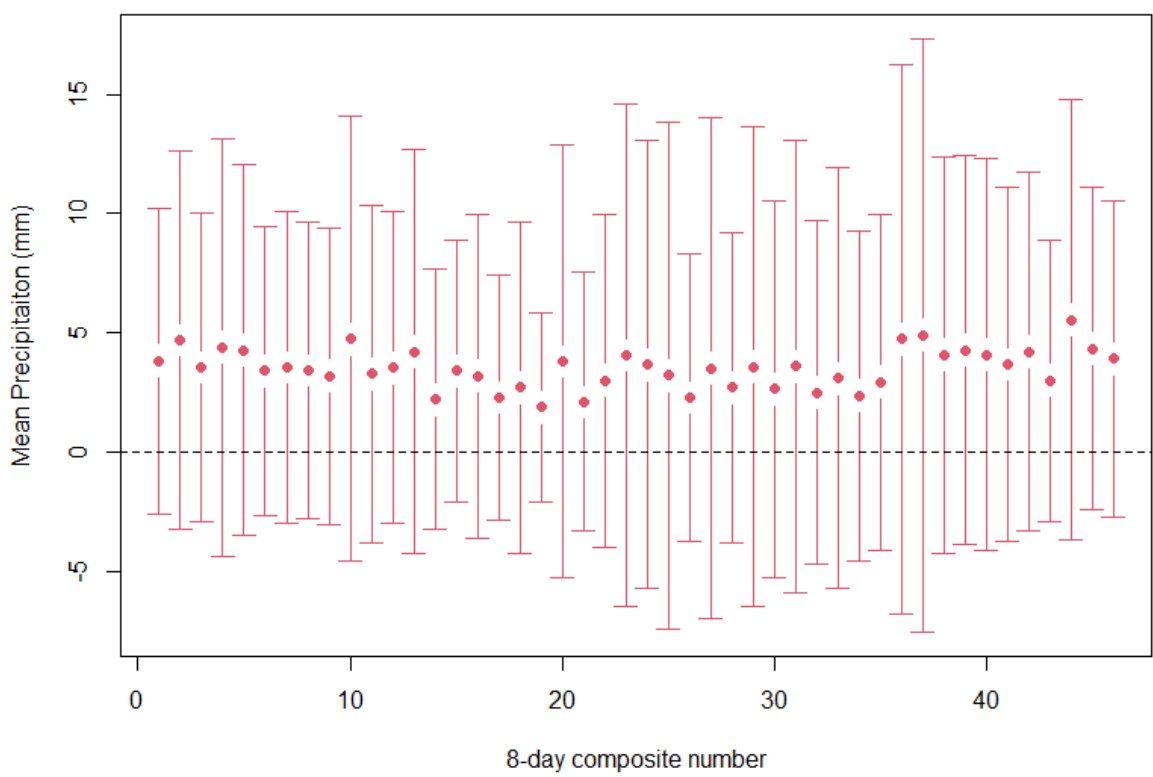

**Figure B1.** Eight-day composite climatology of precipitation sable Island meteorological station (solid circles) . Vertical bar indicates one standard deviation.

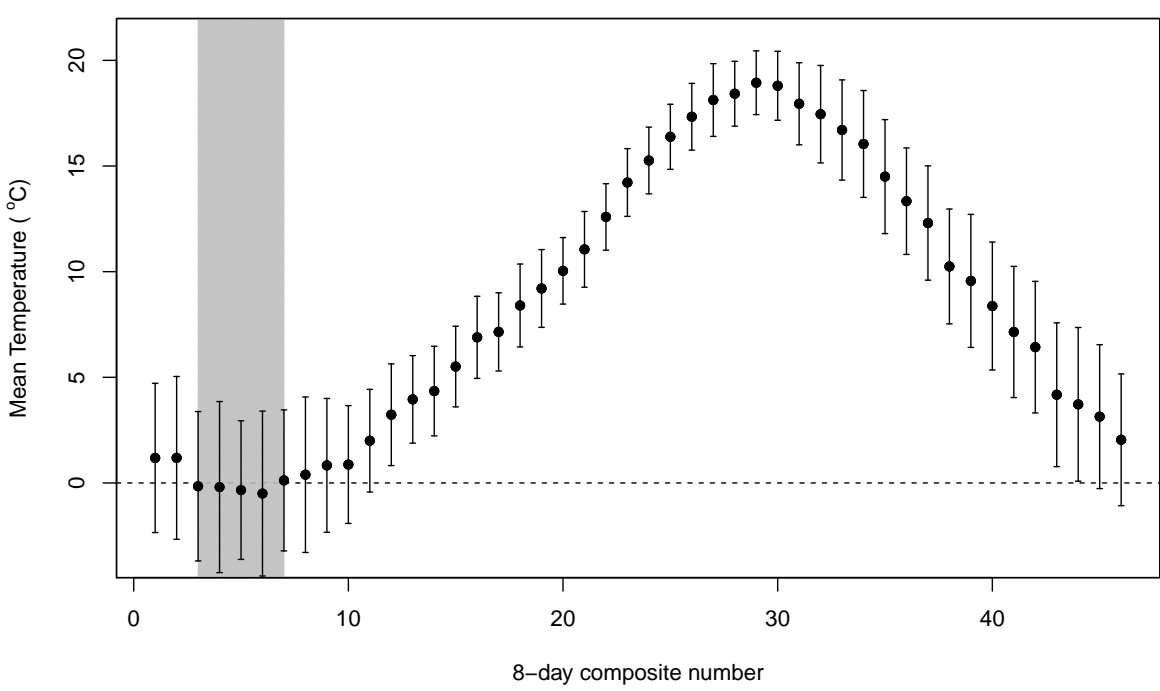

**Figure B2.** Eight-day composite climatology of air temperature over sable Island (solid circles).Vertical bar indicate one standard deviation.