# Peer review of "Enhanced chlorophyll-*a* concentration in the wake of Sable Island, eastern Canada, revealed by two decades of satellite observations: a response to grey seal population dynamics?"

_Biogeosciences, 2021_

## Author Comment (AC2)

Dear Editor and Reviewers,

We would like to thank both reviewers who have provided a detailed review of our manuscript and provided valuable comments and suggestions, which have helped us improving our manuscript. We have done our best to account for all the comments and have responded individually to each of them below (our responses are written in blue for clarity). The major changes of the manuscript include:

- A reorganization and simplification of the ms (e.g., moved some text from the results and discussion to the method section, removed redundant sentences)
- We have toned down the statement about the N-fertilisation contribution to the chlorophyll-a standing stocks in relation to climate change
- We have updated the Seal population model from Hammil et al. 2017 to Rossi et al. 2021 as the new model was published while our manuscript was under review. Note that new model has marginally changed our results and claims. The main change is that we did not need to extrapolate seal abundance of Age1+ for 2017 and 2018

We hope that our response and modification to the manuscript will satisfy the reviewers and editor so that our manuscript will be recommended for publication.

Best,

Emmanuel

REVIEWER #1

The present paper investigates the decadal increase of Chl around Sable Island (on the Scotian Shelf in eastern Canada) related with an increase in the island grey seal population. It addresses interactions between biological, chemical, and physical processes associated to an island mass effect and fits perfectly well within the scope of BG. The increase in grey seal population to explain the positive trend in phytoplankton biomass is a particularly interesting way to investigate. The inclusion of additional optical parameters is also very valuable, as well as the seal haul modeling and evaluation of Chl standing stock which allow an in-depth investigation of the hypothesis that this island fertilization and Chl trends are induced by the seal populations. This study deserves to be published after the consideration of the following points. The major point is that the result section needs to be restructured as there are many back and forth in the writing. The hypothesis and informations provided by using adg and bbp, and how their change can be interpreted regarding chla, clearly need to be stated in the introduction. I would also strongly suggest the authors to moderate their statement about the fact that the increase in Seal population is THE explanation for the increase in Chl as 1) figure 8 and 9 are not convincing and rather suggest the strong influence of complementary forcing and 2) few/no other

dynamic processes specifically related to the IME have been studied in depth in this region. Although the increase in seal populations is a very plausible and convincing part of the explanation, other processes with a significant impact in the decadal Chl increase seem to be at stake. These statements need to be mitigated.

We sincerely thank the reviewer for the supportive comments and suggestions about re-organising the manuscript and moderating our statements. We have taken great care to address the comments to the best of our ability.

In the abstract (Line 14, 24…) and later along the draft there is a confusion when using the terminology "island mass effect (IME)". Here it sounds as the processes inducing the Chl increase. However according to literature (Doty and Ogury, 1956 for instance), IME refers as the biological enhancement itself, induced by the presence of islands. So, please correct that point along the draft.

Thank you for the comments, we have corrected the interpretation of IME

Lines 63,81, 225, 238, 268 and after: use "chl-a"

We have made the modification. Note that "chl-a" refers to the concentration of chlorophyll-a, when discussing the standing stocks of chlorophyll-a, we kept the term "chlorophyll-a".

Check the intermittent use of present vs past tense

Tense of verbs has been carefully checked.

Figure 2, B1 and B2: please modify the x-axis from week number to the corresponding month. A) it will be easier for the reader to follow (especially as it is not usual weeks but 8-days values) B) it will be consistent with the time axes of the other figures.

X-axis of figures B1 and B2 have been modified to corresponding month

**Abstract**:

Line 3:  remove "a detailed". Only part of the processes likely involved in the Chl increase is investigated here.

Done

Line 4-5: Please, shortly explain the interest of using bbp and adg.

Text added

Line 5: replace "the possible mechanisms" by "some possible…"

Text replaced

Line 6: Please precise "8-day climatology" (and hereafter in the manuscript)

Done

Line 23: I wouldn't mentioned "a top-down control" as it is about nutrient supply (so a bottom up process) even though it originates from seal.

Text has been deleted

Line 24-25 and in the conclusion: the authors can't assert "Our findings challenge the idea that the IME is restricted to islands with strong bathymetric slope located in oligotrophic waters of mid-latitudes and tropics, and demonstrate that enhanced marine production can occur in other oceanic regions " as there is an extensive literature of IME taking place in mesotrophic environment such as in the Kerguelen /Crozet islands, the Marquesas islands, the Galapagos, the Gilbert Islands .

Thank you for this pertinent comment, we have modified the text to acknowledge that most studies mentioned above took place in oceanic basin while our study took place on a continental shelf with a much shallower surrounding bathymetry (i.e., mean of 60m) than in previous studies.

**Introduction**:

Line 30: cf above about the def of IME

The words "due to" have been replaced as "known as" to account for the reviewer's comment.

Line 31-2:  please change  "results from …. Or land drainage" to "results from a large range of processes among which…". There are other processes such as atmospheric deposition, human activity inputs….

Change has been made

Line 36-37: and in mesotrophic environment …

Text added

Line 52: how marine mammals can supply nutrients through atmospheric deposition?

According to Theobald et al. (2006), as quoted in their study: "Large colonies of wild animals can emit significant quantities of ammonia (NH$_3$) into the atmosphere as a result of the decomposition of excreta and other waste products", which is deposited downwind of the colonies in the adjacent waters.

Line 58: same comment as Line 6 in the abstract

The term "8-day" has been added before "climatologies"

Line 61 :  "to examine SOME of the possible mechanisms".

Text added

Line 63-65: the explanation about why using bbp and adg (and how interpret their change according to chl-a) should be provided earlier after the first[t] sentence of this paragraph, when DOM and backscattering are mentioned (and should be added in the abstract)

We have added text to the abstract and modified the introduction to explain the use of bbp and adg in the first sentence of the paragraph where these two terms appear.

Line 64: define DOM.

Acronym has been replaced by name

2.2 Satellite and environmental datasets:

L 113: "large cocollithophore blooms occasionnaly.. " Is there any reference to this? Please, shortly explain why you want to remove it from your analyze. Also, the sentence after is not clear.

We found one peer-reviewed article about the significant coccolithophore blooms that occurred in 2003 and 2010: Timothy S. Moore, Mark D. Dowell, Bryan A. Franz. (2012). Detection of coccolithophore blooms in ocean color satellite imagery: A generalized approach for use with multiple sensors. Remote Sensing of Environment, 117, 249-263. doi:10.1016/j.rse.2011.10.001, as well as media release, see for instance: https://www.iirs.gov.in/Sable-Island, https://www.dal.ca/news/2012/10/19/unlocking-sable-islands-scientific-secrets.html and https://visibleearth.nasa.gov/images/67082/phytoplankton-bloom-off-nova-scotia

We removed these bbp data from the time series, as it was much higher than the other data in the bbp climatology and time series, and it impacted all further calculations by greatly altering the mean (e.g. in 2010, surrounding Sable Island bbp exceeded 0.03 m^-1; the second-highest bbp value in the time series is below 0.01). We removed full images rather than using a threshold in order to compare the same imagery between regions. As the corresponding adg and chl-a were not significantly impacted, we decided to leave these data in.

2.3 Climatology…

Section title: add "8day "climatology

Title of section has been updated

Line 129: "SI"

Text corrected

Figure 2: In the legend: correct "SB" with "SI"

Legend updated

Overall, it is extremely difficult to make the link between the figure and the text. See complementary comments below in section 3.1

Thank you for the comments, we have made an effort to better evidence the link between the text and the figures. We have added references to the figure in the text when deemed appropriate

Line 148: What do you mean by "to obtain detailed maps of the chl-a in the SOM5"? do you mean THE SOM5 chla map (not a plurial) in Figure 3?

We have removed the term "detailed maps of chl-a" as it did not accurately express our thoughts and updated the text.

Line 156-8: Is the 2011 unusual bloom removed from all the manuscript analysis or just for the 5-year trends analysis? If it is only removed for the trend analysis, how does it influence the interpretation with/from the other analysis?

The unusual blooms that occurred in 2003 and 2010 were removed from all the study analysis, including the SOM and time series analysis.

2.4 Chlorophyll-a concentrations….

Line 185: add ")" after Figure 3

Closing bracket added

While maps in (a) and histogram in (b) are clearly presented and can be easily understood as their interpretation is close to those of the usual clustering or EOF, this is not the case of the grey path. Thus, please further explain how to understand/interpret the path and the location of the numbers along it.

We have added explanation to the legend in Figure 3 and we have also updated the Figure 3 by replacing the month number by the month abbreviation as we think it will avoid confusion between node number and month of the year.

2.5.1 Decadal change in grey seal abundance…

I went through the publication of Hammill et al 2017 and den Heyer et al. 2020. Finally, it is unclear what numbers in the present paper in Table A1 come from observations or are derived from modeling. Please clarify.

While our manuscript was under review, a new seal model was published. The seal abundance represents age 1+ seals and comes from the model which is fit to pup counts to estimates annual abundance. The new seal abundance, and all the derived results have been updated in the study,

without significant change to the findings. Note that the number of seals is now slight higher than in the original study. Table 1 represent 5-year average of annual seal abundance

If the seal trend is derived from modelling and follows a "regular " curve as in Hammill et al 2017 , I guess it explains why there was no interest in showing a trend derived from annual average instead of the 5 year mean. It may deserve to be mentioned.

We have added text to reflect the reviewer's comment. Note that Line 417 of the manuscript, we discuss the annual increase of seal, which is of 5% as described in Rossi et al. 2021.

Line 213-Equation 1: why illustrating the equation with P4 instead of P1 which would be more logical?

 We have used parameters rather than number of seal from a given population to illustrate the Seal Model and the parameters for each period are summarized in table A1. The period used does not change the shape of the curve, only the abundance of seals such that any period could have been selected.

2.5.2 From seal population…

Line 233-234: "The release of N…" What does this sentence provide? Rather move in the discussion section if relevant (for instance line 459)

We thank the reviewer for pointing this, we have moved the sentence to the discussion.

Line 238-239: " chl-a multiply by the SOM5 area" : you mean the surface plume around the island delimited by the 1 mg.m-3 isocontour? please clarify

We have updated the text in the method section (Lines 152 and 197) to provide an accurate definition of the SOM5 plume used in the rest of the study.

Line 241: what is "HL2"?

HL02 is an oceanographic station located  to the south-east of Halifax, Nova Scotia. We removed the term "HL02" from the main text. Readers can refer to Casault et al. 2020 to have additional information on the station.

3.1 Evidence of enhanced chl-a…

The first paragraph should be moved to the introduction. It does not present ongoing results.

Overall, this section is very hard to follow, because a) showing the time series of all the boxes makes impossible to distinguish the results from each other (except if you can spend 20 minutes on it). This figure needs to be rethought, perhaps by averaging some boxes together showing the std around them, b) the text back and forth from average pattern in optical properties to boxes patterns, then average patterns again. It is the same for the hydrodynamic section. It is about the Labrador current Line 299 to 301, then KE (by the way, why talking about KE as you don't use this information later), then back to the Labrador current Line 303. The main patterns then the boxes again.

At the end of this section it is extremely hard to remember/understand what are the take-home messages. This section needs to be rewritten and clarified.

Thank you for the comments, we have modified the figure 3 to make it clearer by regrouping the small boxes that have similar patterns. We have also rewritten this section to emphasize the message and have a better flow for the reader. We have added subsections for the control boxes (the 3 large boxes) and for the small boxes that help with identifying the IME. In fact, we used the small boxes to demonstrate the IME leeward of Sable Island, as the SI box initially chosen did not demonstrate a chl-a or seasonal patterns different from the two control boxes. This is explained by the fact that averaging satellite-derived chl-a data in a large box subjectively chosen smoothed the signal. To evidence a possible IME, in an objective manner, without a priori knowledge of the area impacted by it, we decided to compute the signal in small boxes as to identify where the IME occurs. We did that exercise for the 3 bio-optical properties to ensure that the chl-a signal observed was due to biological production and not contamination by other components such as dissolved organic matter. Once the IME was proven, we used a SOM approach to have a better idea of its extent, but also to look at its change with time (season and decades).

Added comments:

Line 272-275: it is written that the NE box exhibits the highest chl-a while Figure 2 shows equivalent or stronger bloom in 5 boxes around Sable island.

Here we meant that NE has the highest Chl-a of the 3 control boxes, we have modified the text to reflect this.

Line 283: it is stated ". Backscattering magnitude is related to abundance and inversely related to particle size (Slade and Boss, 2015) and here the timing of maximum backscattering is consistent with the time of year when large phytoplankton such as diatoms and dinoflagellates reach their minimum abundance, while small flagellates are most abundant." In the abstract Line 64 you mention bbp to "indicates the presence of mineral particles due to resuspension", as in section 3.2.1. Therefore, what is the causal link here with the dinoflagellates?

We have added text to the statement. In fact, in deep (> 100m) open waters, away from terrigenous inputs, backscattering is driven by phytoplankton and not mineral particles, such that changes in bbp can be explain by changes in phytoplankton cell size distribution and abundance. However, we have removed the statement as we have no evidence that the backscattering coefficient is driven by phytoplankton only.

Line 295: ".. without obvious enhanced chl-a" I didn't understand this part of the sentence as Fig 2 shows chl-a is enhanced compared to SW box, and even to NE box for boxes 8-12. What did you mean?

Thank you for the comments, we have updated the text, and in particular the statement "for the remainder of the year", as a clear increase of chl-a, as pointed out by the reviewer, is visible early in mid-to late fall.

Line 299-301: the depth-averaged Labrator current is provided and this value is used to say that there is a decrease close to SI, but it is compared with surface value. Why don't you use the same consistent

data set (ie OSCAR) to illustrate the current weakening? Also, as it would be useful at least in the appendix to illustrate that point with a seasonal map.

We have modified the text to emphasize the OSCAR dataset, and added maps in the SI, while we have also kept the information on the kinetic energy from Brickman et al. 2012 as it provides information on mixing.

Line 308-9: "However …" you can't completely discard this hypothesis, as finer scale dynamics than those modeled in Zhai et al 2011 could be at stake, especially considering the small size of SI and the complicated bathymetry and induced dynamics on the shelf.

Thank you for the comment, we discarded the statement.

Lines 310-14: the cause and effect links between the different sentences are unclear.

We agree with the reviewer and we have simplified the sentence and moved it to the previous paragraph.

3.2 Timing of biomass.

Title and Line 331: this is chl-a, not biomass

Title has been updated

Line 317 to 321: " Self-organizing …. 1999)" should be removed/moved to the SOM methodological section.

We have moved the text to the method section.

Line 324: be more explicit about the fact that "The largest extent of the plume when defined with a 1 mg.m-3 contour occurred ….." It need to be state asap as with a higher magnitude contour the plume would have been highlighted by node 7, corresponding to the spring bloom.

We believe that the threshold used is independent of the node and the season as the SOM approach evidence a plume of high chl-a around SI that occurs in node #5. Outside of the spring bloom period (i.e., node #7 as pointed out by the reviewer), the largest extent of the plume occurs in Winter (i.e., node #5) with chl-a that are higher that the surroundings. Therefore, changing the threshold for this node, would just change the surface area of the plume but would not change the interpretation of the node #5 (for instance, selecting #7).

Line 332: "the hypothesis of the IME" see above about the def of the IME

Thank you for the comment, we have corrected the text.

Line 335: not clear

We have updated the sentence and made it clearer.

Table 1&2: should appear after being cited in the text. Also, in the legend, it is stated that it is a linear regression of annual chl-a. Therefore, what does it mean "vs time in years"?

A linear regression is done on a dependent variable Y against the dependent variable X, on our case, the dependent variable is time expressed in year, we think that it is accurate to cite both variables.

Line 336: "the annual seasonal trend" is it annual or seasonal? Unclear

We have updated the text to make the term clear.

Line 336: "calculated foe the SW and NE and SOM5" what do you mean by SOM5? Average within the area delimited by the isocontour= 1 mg.m-3 on SOM5 map? Please clarify

The definition of the SOM5 region is given in the method section, Lines 152 and 197 of the updated manuscript.

In this paragraph, please gather results about adg first, then bbp. It goes back and forth and it is hard to follow.

Thank you for the comment, we have modified the text to group the discussion on each property

Line 344: You propose an explanation of the adg increase, and what about for bbp?

What is also unclear is why a significant positive trend in adg while chl-a doesn't show a trend suggests that chl is not contaminated by adg? I would have rather expected that showing the same trend (or no trend) between the 2 parameters would means no contamination.

When using band ratio algorithms, such as OC3, the presence of adg, which does not covary with phytoplankton concentration (i.e., case 2 waters) will artificially increase satellite derived chl-a due to its high absorption in the blue channel (443/490), which mimic phytoplankton absorption (see for instance Darecki and Stramski 2004, Remote Sensing of Environment). If both properties would increase in tandem, it would be difficult to conclude that the increase in chl-a would be reflecting an increase in phytoplankton biomass or just contamination of the ocean colour signal by adg.

Is the section related to Table 1 and the annual trend really noteworthy? It may be more impacting to directly present the seasonal trend after the SOM section which also provides seasonal information's to simplify the take-home message.

We have removed Table 1 and the text about the annual trend as suggested by the reviewer to simplify the text and get directly to the seasonal trends.

Line 345 "in winter" (no THE between)

Text corrected

Line 348-50: what does the adg high increase for both NE and SOM5 region suggest?

While in the SOM5 the increase could likely be due to the increase in N-release by seals, in particular close to the islands, it is difficult to elaborate for the NE box without an in-depth analysis, which is out of scope of the study. We have added a sentence to state that explanations about the increase in adg in NEb is out of scope of the study and would require further study.

Lines 351-53: is it necessary to describe everything from the figure. Rather focus on the take home message which is diluted. What do we need to remember in winter from the chla, adg and bbp relationships, and what do we learn from the SOM5 vs control boxes.

We do think that every panels needs to be discussed if it is included in the figure. The figure provide a visual that supports Table 1 (previously Table 2). The figure shows both the trends and the magnitude of Chl-a and Adg in the winter, particularly the very high winter Chl-a at SOM5, and also emphasizes the departure from these trends seen in bbp. The magnitude of bbp is high for SOM5, but not significantly changing. The figure also shows that Chl-a at SOM5 is often higher than the spring bloom.

Figure 5 should be shown in this section rather than section 3.2.1. By the way there is a misfit between the text which relates to chl standing stock (defined in section 2.5.2), while Fig 5 is chl-a purely.

We have moved the figure to the proper section. We agree with the reviewer about the chl-a standing stocks. These values represent a single number that integrate the surface area and depth. We think that it is important to show the figure in chl-a (mg m-3) to emphasize the increase in size and chl-a concentration of the plume. We have moved the reference to Figure 5 earlier in the sentence when introducing the averaging of chl-a

3.2.1 Simultaneous increase in …

Explanations/hypothesis from lines 364 to 381 should be provided adequately before, in sections 3.1 and 3.2. It would again avoid the need to go back and forth which would fluidify the reading and help understand earlier the relationships/hypothesis between chla vs adg and bbp

We have moved the text to section 3.1 and 3.2 to increase the fluidity of the manuscript as suggested by the reviewer.

Line 376-78: hyp 2 is discarded, but then to what the adg increase is related to?

This is a good point, we can only speculate that the increase in adg would be related to 1) the increase in chl a, 2) increase of transport of detritus/gelbstoff. The causes of the increase in gelbstoff on the scotia shelf is not the scope of the study.

Line 370: "shoaling" of what? Of the mixed layer depth? Of nutrients? Unclear

We meant shoaling of nutrients, text has been added.

Line 376: what is "slope waters"? you mean uplift of isopycnes when the flow encounters the island? Vertical mixing from what? Seasonal convection? Horizontal mixing or advection (from where)? Please clarify

The slope waters designate the water mass that is located on the continental slope and advected on the Scotian Shelf from the deep basin, we have added text and references to clarify this.

Trying to synthetize: the winter time Chl increase can be related to biomass increase thanks to nutrient supply a) from the ocean (induced by the ocean dynamics or seals), and/or b) from land (which can be detected by adg), and/or c) related to particle resuspension (related to bbp).

Yes, this is the hypothesis we have stated and confirmed/inferred using the satellite time series.

You can't definitely discard potential impact of changes in ocean dynamics as they are never explicitly investigated (and this is not the topic of the paper). First, regional dynamics can be partly responsible for SI Chl-a increase as an increase is also observed in the control boxes. Local processes could even enhancement nutrient uplift (for instance stronger currents could uplift more nutrient from the bottom), stronger EKE could have the same impact. What about nutrient advection from the Gully MPA? Please, be less affirmative. Investigating the seal population is very interesting, you do not need to overconclude about the physical processes.

Thank you for pointing this out. We have toned down our conclusion and added text to list possible other mechanisms. We acknowledge that there is an overall increase of chl-a in winter on the Scotian Shelf, although the mechanisms are not fully understood, however, what is striking is the rate of increase of chl-a in the plume located leeward of SI, which is twice as high as the surrounding area. For this reason, we do think that the increase of seal population and associated N fertilization contribute to that increase at a level of XX percents.

We doubt that the increase in nutrients around SI would be due to nutrient advection from the Gully MPA as Strain & Yeats (2005, Nutrients in the gully, Scotian shelf, Canada, Atmosphere-Ocean, 43:2, 145-161, DOI: 10.3137/ao.430203) showed that, while nutrients in the Gully are higher than in their surroundings, they remained trapped within the canyon with limited overflow.

Line 382-4: if there is a continuous decrease in nitrate since 2012 how is the Chl increase in the control boxes can be explained?

Nutrients are not a limiting factor on the Scotian Shelf, only during the spring bloom such that high biomass cannot be sustained all year long. Despite an overall decrease in nutrients, chl-a has not been decreasing but we are observing changes in community structure with an increase in the abundance of flagellates (manuscript in preparation).

Line 388-396, and last sentence: move/adapt to the introduction section. Here, directly present your own results and gather with section 3.2.2 (finally, section 3.2.1 as presented here would be removed)

Thank you for the suggestion, we have included some of the text in the introduction section and we have removed section 3.2.1.

3.2.2 Seal abundance…

Line 403-420: in the text it is refered to as "normalized chla" while in Fig 7 legend it is Chla, which can even be interpreted as standing stock. Please homogenize.

We use the "Normalised chl-a" term as we focused on the seasonal cycle of chl-a independently of the concentration to simplify the figure. We have now changed the figure and added a scale on the right side of the figure for chl-a in the NE such that the term "normalized" can be removed.

Here again there are back and forth: you start with changes in P1 and P4, then compare with NE box, then go back to results in SOM5 in agreement with decadal trend, talk about seal and go back againto SOM5 results.

Thank you for suggesting the change to improve the clarity of the manuscript. We have updated the paragraph to group the discussion on each variable/period.

Figure 8 should be in the text and not at the end (ditto later for Fig 9)

We have change the position of the Figure 8 compare to the text. Note that the final position of the figure relative to the text will be determine in the final version of the manuscript (proofread) with a two columns lay out (see biogeoscences article format)

Part of the text at the beginning of page 19 should be moved/combined to the methodological section. He would help to focus on the main results. This paragraph (more than a page is far too long).

We have removed some of the text to avoid redundancy with the method section and added some of the information in the method. The paragraph has been shortened to only focus on the results.

Line 433: what are the significance associated with the pearson coef?

We have added the p-value associated with the correlation coefficient. For r2 = 0.54 we found a p-value of 2.4e-4 and for r2 = 0.86 we computed a p-value of 5.9e-9

Line 435: I don't understand how it can be concluded that half of the Chl standing stock increase between P1 and P4 comes from the seal N-release according to Figure 8 where the model is far far below the satellite derived Chl standing stock.

When the difference between P4 and P1 is calculated for the satellite observations, several values are negative, such that the annual sum is lowered, for a total of 757 tonnes, while the value for the model over the entire year is 311 tonnes or about half. However, when focusing only on the winter period (i.e., breeding season) the contribution by seal fertilization is about 20%, perhaps in better visual agreement with the figure. We have updated the text to reflect this.

Line 438: "could support PART (OR HALF) OF the observed plume

We have re-written this paragraph, including this sentence to account for the reviewer's comment above.

Line 450: go to line after "necessary"

Done

3.2.3 Synchronized decadal…

This section is not very convincing. Fig 9 shows a far stronger trend of climate forcing than seal fertilization while in the text it is highlighted:" This is another striking finding that advocates for the impact of seal fertilization of the nearby ocean as no other mechanisms have supported the strong increase of chl-a in this region ». Not finding the explanation (as once again physical processes haven't been investigated in depth) does mean that seal fertilization is the main cause. Figure 8 and 9 rather suggest that seal abundance is part of the explanation, not THE explanation

We agree with the reviewer that our message was overstated and we have moderated it to conclude that the contribution by seal to the chl-a increase was non-negligible but not the main cause.

What about the significance of the correlation coefficient Line 491, calculated with 4 points (if related to Fig 9).

Thank you for noting this, the p-value for this relationship is 0.1, which is therefore unconclusive. We have added the p-value to the text.

Conclusion

Line 502 and 525: see one of my first remarks about mesotrophic IME in literature

We have modified the text: we have deleted the statement about the novelty of finding IMW in mesotrophic environment.

Line 505-506: provide the causal link when you state that the there is also a winter increase in adg and bbp as in chl.

Given that our study is restricted to satellite data, it is challenging to elaborate on the processes that explain the increase in adg and bbp, while the increase in Seal feces could explain the increase in adg, we cannot not assert it without in situ measurements. This is the same for the increase in bbp. At this stage we could only speculate.

First paragraph: do not repeat the factual results on chla, adg and bbp trends. Rather report the take home message about seasonal increase over the last decades and to what it can be (or not) related to by investigating adg and bbp in parallel

Paragraph 2: here again, this is not because adg and bbp didn't explained the chl increase that "the only drastic change that has occurred is the increase in seal population"

We have toned done our statement as suggested by the reviewer.

Last paragraph: you state line 529 that the N release can support the chla standing stock of 254 tonnes in winter. Not convincing from Fig 8 and 9. This statement need to be mitigated. It is a convincing PART of the explanation

Figure 8 shows the difference between P1 and P4 such that is reduces the contribution from seal fertilization. If we account for P4 alone, the possible chla standing stocks is therefore 254 tonnes. If have emphasize that the contribution by seals in only a part of the explanation to the chl-a increase.

**Commented [AH1]:** Is this "It has emphasized"?
* * *
REVIEWER #2

**General appraisal**

The paper by Devred and co-workers presents an attempt to quantify the role that an increasing marine mammal population (grey seals in this case) can have on water quality parameters such as the chlorophyll concentration, through the release of nitrogen in coastal waters. The study area is Sable Island (SI) on the Scotian Shelf in eastern Canada.

For this purpose, they use satellite ocean colour-derived chlorophyll and inherent optical properties, namely the particulate backscattering coefficient at 443 nm, $b_{bp}(443)$ and the coloured dissolved organic matter plus detrital matter absorption coefficient at the same wavelength, $a_{dg}(443)$).

The topic is relevant for publication in BGS.

Overall this is a rather interesting paper. As the authors say (lines 23-24), the absence of in-situ data to confirm the results, which are entirely based on satellite remote sensing products, is a limitation but, certainly not a significant flaw that would prevent publication of their work.

The study site and methods are rather well described, although the rationale for using self-organising maps (SOM) and their presentation are not that clear.

Thank you for your comments, we have made substantial improvements to the manuscript following the reviewers comments/suggestions with the goal of improving the fluidity and simplifying the manuscript

I did not notice much that would require clarification here, except maybe to know whether seal-generated nutrients are rather produced on land and then have to be washed out to the sea to have an impact there or a significant part of it is directly generated at sea. Looks a bit trivial but might have a significant impact on whether this nitrogen added in the ecosystem can indeed have an impact on phytoplankton growth (what's written in section 2.5.2 seems to assume all is produced on land).

It is very possible that some of seal excretion occurs at sea, however, most of it is assumed to happen on the island and is washed out at sea, this is the reason why we looked at meteorological data (precipitation and temperature) and believe that a lag between the peak of seal N fertilization and phytoplankton response exists.

The paper is however quite lengthy and could be shortened significantly. The message that the authors want to convey is somewhat lost. A reorganisation is needed as well, with bits of method-like descriptions to be removed from the results/discussion section. Overall, the methodology has to be clarified. This paper is hard to read in particular because the Figures do not make a very good job in conveying the important results (quite often hard to locate/identify on Figures what we read in the results/discussion in particular).

We have made an extensive effort to simplify and reorganize the manuscript as suggested by both reviewer. We thank the reviewer for this constructive comment that will make the manuscript easier to read.

I also found that analysing the results based on both 12 arbitrarily located boxes and 9 SOM-derived regions is somewhat confusing. At least I could not find a clear justification for doing so. Looks like using only the latter might make more sense.

We thank the reviewer for the comments. However, we took an objective approach that could demonstrate that the IME exists and we believe that by determining first an area or a threshold for chlorophyll-a concentration might bias the study. We found that delineating small boxes around sable island without a priori knowledge of the phenomenon would be the best, most sound approach. We updated the approach following reviewer's #1 comment to make easier to follow and we have emphasized the main message.

**Detailed comments**

- Figure 1: I did not understand what the boxes labelled B1 to B12 were until I reached section 3.1, but then I wonder why the seasonal cycles in Fig. 2 are displayed for these arbitrarily defined 12 "square" boxes, instead of being displayed for the 9 regions that the SOM has identified?

  The SOM analysis provide the spatio temporal distribution of chlorophyll-a on the Scotian Shelf while the small boxes analysis focuses on the seasonal cycle of chl-a, adg and bbp

in the vicinity of Sable Island to demonstrate the IME. These are two different approaches that do not have the same aim.

- Figure 3: not sure I understand this one, in particular: "The centroid of each month in SOM space is shown as the dark grey path, starting with January (1) near the centre, proceeding counter-clockwise to December (12)". And what is the chlorophyll concentration displayed in each of the 9 panels in (a)? Annual average? Why not rather display a map showing the spatial distribution of the 9 phenotypes (I guess one can call the 9 seasonal cycles shown on panel (b) "phenotypes"?). That would clearly show where the different seasonal patterns occur. But are panels in (b) actually showing seasonal cycles? Sorry but I realise I am actually confused by this SOM analysis. And, still on this figure: if the spatial patterns are important, then the maps should be much bigger.

  Following the reviewer #1 suggestions, we have added explanations to the Figure 3 legend. The SOM analysis is similar to a cluster analysis where images with similar features (i.e., spatial distribution and magnitude of chlorophyll-a concentration) are grouped together, the final result is 9 maps (i.e., clusters of similar images) that show the main distribution of the chl-a in the area of interest. The right panel (-b, histogram) shows the frequency of occurrence of each pattern. In this manner we were able to show that the pattern number 5, which show a plume of chl-a leeward of Sable Island occurs mainly in Winter, which corresponds to the seal breeding season. We have added text to clarify this.

- Section 3.1, lines 282 to 290: you cannot explain an increase in $b_{bp}$(443) by a factor of 2 to 3 when chlorophyll does not change at all by a change in the phytoplankton size only. The only reasons I can see here to explain the huge increase in $b_{bp}$(443) from week 17 to 25 in all 12 boxes (when Chl is steadily around 0.5 mg m$^{-3}$ in all boxes) would be coccolithophorids or mineral particles (sediments). Therefore, it seems that the elimination of these events is not actually completely performed by what you describe in section 2.2.

  Thank you for the comment, we agree that phytoplankton community structure alone cannot explain the large increase in $b_{bp}$(443). We have removed large coccolithophore blooms that have been documented in the literature (see response to reviewer #1) but we cannot remove smaller coccolithophore blooms or resuspension events that do not have a strong signature.

[Figure]

The figure above shows the mean time series of $b_{bp}(443)$ for the 3 large boxes, where all years are plotted on top of one another (the x-axis is the Julian day of the year, and y-axis is $b_{bp}(443)$). 2 large coccolithophore blooms are highlighted: the orange solid line corresponds to 2003 and the red line corresponds to 2010, seen to occurr in the SW box and the SI box. These have been removed from the analysis. You can see that the NE box has other events of note, but we removed the images occurring during the 2003 and 2010 events that would have significantly impacted the statistics for the SW and SI boxes.

- Beginning of section 3.2. This is stuff for the method section, not for discussing results. Considering my comment above, I cannot really comment on Section 3.2

  We have reorganized the manuscript and the text at the beginning of section 3.2 has been moved to the method section.

- Line 289: I think an increase from 100,000 to 300,000 is a 200% increase, not 300%. (100 x (300,000 – 100,000)/100,000).

  We have correct the percentage from 300 to 200%

- Lines 385-400: Does not a lot of this already appear in the method section?

  Yes, a lot of the material in this paragraph has already appeared in the method section. This was a reminder, however, we have removed the text.

- Legend of Fig. 7: what do you mean by "images"?

  We have replaced the term "images" by "mean.

- Lines 420-421: well, instead of telling us that the "mathematical formulation of the model was rather arbitrary" you could tell us what the model is. Would definitely be more useful.

  We have modified the text to explain the model, thank you for the comment

- Not sure why Figs. 8 and 9 are not included in the main text, like the others.

  We used the Biogeosciences latex format. The position of the figures will be determined for the final version before publication. We will ensure that the figures are appropriately located.

- Frankly, all page 19 is really hard to follow. There is too much in there, without a clear message on what you want to tell us. That section is where you definitively lost me.

  Reviewer #1 has made a very similar comment, we have modified and shortened page 19.

- Line 466: you may have shown a correlation but, claiming that you have identified a "causal link" is probably a bit of a stretch here. You cannot say this.

We have toned down our statement.

- Figure 9 is another one that is not that easy to understand. The values on the right scale are about ten times lower than those we read on the left scale, so that I think the right scale should read "Change in the Chl-a standing stock", right?

Thank you for the reviewer to point this out, we have updated the legend of the right scale.

- Line 479 (related to Fig 9): I do not see these numbers on Fig. 9. The final value in P4 is close to 3,000 actually.

The values P1N, P2N, P3N and P4N are not shown on Fig. 9. We have added these to Fig. 9.

- Line 526: does anyone really said that the island effect is only supposed to occur for oligotrophic waters? References would be good to have here.
- We did not find in the literature a study about IME that occurred on a continental shelf, most of the studies focused on island located in oceanic basin surrounded by oligotrophic waters, however, we decided to remove the sentence to avoid confusion.
- Appendices A and B are not called in the main text. Appendix A could actually be incorporated in the method section, and appendix B does not seem that useful.

  We have left the SI in its current form as we believe that it contains interesting information that can be consulted at the discretion of the reader.

---

## Referee Report (RR1)

I thank the authors for taking care of my previous comments. I found the paper much easier to read and follow. I recommend the publication of this paper after considering minor comments:

L162 to 169: this paragraph may be moved at the end of section 2.5.2 where the calculation of Chl standing stock is defined and where it is explained why P1 is independent of possible seal influence

Line 164: why do you use the SW box to derive the slope of Chl-a whereas in Line 417 and in Fig 7 this is the NE box which is used? It is maybe a typo.

L281: as you developed a little bit your interpretation for adg Line 276-277 (which was great! Please keep it). The same kind of explanation for bbp would be helpful

L 331: Can you provide some possible interpretations of the causal links of adg and bbp trends with those of Chl during wintertime?

Figure 9: the left y-axis could be more extended upward to reach approximately the highest values. You may also consider to adapt the right y-axis, as it was in the initial version. Right now, it looks like the seal fertilization induced Chl seems stable, the increase is not visible

Conclusion section:
L 449: it is written that "no trends in particulate bbp and adg were markedly different from the surrounding environment". Not sure to understand… Figure 4 shows that there are trends in the 3 variables during wintertime, which for all of them are stronger around SI than in the SE and NE regions.
Also, in this section the bbp and adg patterns are only descriptive, can you add some causal links of adg and bbp trends with those of Chl.

L 459: replace "strong" by "good"

Typo:
Line 20: "during winter"
Line 30: I am not sure that Signorini et al 1999 deal with benthic resuspension
L 140: climat**olo**gies
L 146: for boxes 1, 2, 3 **and 4** , check with Figure 2 where it is written group 1= boxes 1 to 3
L 147: you could add at the end of the sentence "as illustrated in Figure 2"
L 214: "between males and females the same". I am not sure to understand "the same"
L 267: the shelf show**ed**
L 305: in **w**inter
L 313 in spring
L 322: three "ROIs", please define
L 382: the seal supported the chl-a annual standing stock …
L 418: correspond**ed**
L 423: differences occurr**ed**
L 424: fertilization provid**ed**

L 430: provide a good agreement
L 441: around SI, region referred to as …
L 441: occurr**ed**

---

## Author Response (AR2)

Response to Reviewer:

Dear Editor,

We have completed the minor revisions required by the reviewers #2, in particular, adding text about the casual link to tentatively explain changes in the backscattering coefficient over time. We thank again the reviewer for the detailed comments. Note that we did not use the track change for minor typos highlighted by the reviewer, but we have accounted for all of them.

We hope that the manuscript will be accepted for publication in Biogeosciences.

Best Regards,

Emmanuel Devred

Reviewer #1:

We thank the reviewer for accepting the manuscript for publication as is.

Reviewer #2:

We are grateful to reviewer #2 for providing another set of comments that will help making some last, but not negligible, improvements to the manuscript.

I thank the authors for taking care of my previous comments. I found the paper much easier to read and follow. I recommend the publication of this paper after considering minor comments:

L162 to 169: this paragraph may be moved at the end of section 2.5.2 where the calculation of Chl standing stock is defined and where it is explained why P1 is independent of possible seal influence

We have followed the reviewer recommendation and moved the paragraph between Line162-169 at the end of section 2.5.2

Line 164: why do you use the SW box to derive the slope of Chl-a whereas in Line 417 and in Fig 7 this is the NE box which is used? It is maybe a typo.

In Figure 7, the NE box is showed only as a reference to emphasize the different annual cycle of chl-a in the vicinity of Sable Island compare to the control boxes, the NE box in that case. The use of the SW or the NE box does not provide very different results as the slope for SW and NE in winter are 0.015 and 0.011 respectively

L281: as you developed a little bit your interpretation for adg Line 276-277 (which was great! Please keep it). The same kind of explanation for bbp would be helpful

We have added some explanation to the seasonal variation of bbp

L 331: Can you provide some possible interpretations of the causal links of adg and bbp trends with those of Chl during wintertime?

We have added two sentences to provide possible explanation for the findings. As demonstrated for the seasonal cycle, adg variations seemed to be related to chl-a, while bbp seemed to be driven by hydrodynamic and resuspension of particles (non-living) in the water column. The increase in adg is explained by the increase in chl-a, while the decrease in bbp is supported by changes in physical forcing on the Scotian Shelf and notably an increase in stratification that impedes (Hebert et al. 2021) particle resuspension to reach the surface waters.

Figure 9: the left y-axis could be more extended upward to reach approximately the highest values. You may also consider to adapt the right y-axis, as it was in the initial version. Right now, it looks like the seal fertilization induced Chl seems stable, the increase is not visible

Thank you for the comment, we have updated the figure and adapted the axis to match the initial version of the figure 09

Conclusion section:
L 449: it is written that "no trends in particulate bbp and adg were markedly different from the surrounding environment". Not sure to understand… Figure 4 shows that there are trends in the 3 variables during wintertime, which for all of them are stronger around SI than in the SE and NE regions.
Also, in this section the bbp and adg patterns are only descriptive, can you add some causal links of adg and bbp trends with those of Chl.

The text about absence of trends in an oversight from the original version of the manuscript, in which we presented the annual trends for chl-a, adg and bbp that did not show any significant trends. We have corrected the text and added some explanation about the casual links between the variables.

L 459: replace "strong" by "good"

Done

Typo:
Line 20: "during winter"
Text corrected
Line 30: I am not sure that Signorini et al 1999 deal with benthic resuspension
L 140: climatologies
Text corrected
L 146: for boxes 1, 2, 3 and 4 , check with Figure 2 where it is written group 1= boxes 1 to 3
The legend has been corrected and reflects the text and the manner we carried out the computation
L 147: you could add at the end of the sentence "as illustrated in Figure 2"
Text added
L 214: "between males and females the same". I am not sure to understand "the same"
This text has been deleted
L 267: the shelf showed
Text corrected
L 305: in winter
Text corrected
L 313 in spring
Text corrected
L 322: three "ROIs", please define
Tefinition of the three ROIs has been added
L 382: the seal supported the chl-a annual standing stock …
Text corrected
L 418: corresponded
Text corrected
L 423: differences occurred

Text corrected
L 424: fertilization provided
Text corrected
L 430: provide a good agreement
Text corrected
L 441: around SI, region referred to as …
Text corrected
L 441: occurred
Text corrected